# Contrasting parental roles shape sex differences in poison frog space use but not navigational performance

**Andrius Pašukonis[1,2,3]\*, Shirley Jennifer Serrano-Rojas[3,4], Marie-Therese Fischer[3], Matthias-Claudio Loretto[5,6], Daniel A Shaykevich[3], Bibiana Rojas[7,8], Max Ringler[9,10,11,12], Alexandre B Roland[13], Alejandro Marcillo-Lara[14,15], Eva Ringler[9,16], Camilo Rodríguez[11], Luis A Coloma[15], Lauren A O'Connell[3]\***

[1]Institute of Biosciences, Vilnius University Life Sciences Center, Vilnius, Lithuania; [2]CEFE, Univ Montpellier, Montpellier, France; [3]Department of Biology, Stanford University, Stanford, United States; [4]Universidad Nacional de San Antonio Abad del Cusco, Cusco, Peru; [5]Technical University of Munich, TUM School of Life Sciences, Ecosystem Dynamics and Forest Management, Hans-Carl-von-Carlowitz-Platz, Freising, Germany; [6]Berchtesgaden National Park, Doktorberg, Berchtesgaden, Germany; [7]Department of Interdisciplinary Life Sciences, Konrad Lorenz Institute of Ethology, University of Veterinary Medicine Vienna, Vienna, Austria; [8]Department of Biology and Environmental Science, University of Jyväskylä, Jyväskylä, Finland; [9]Division of Behavioural Ecology, Institute of Ecology and Evolution, University of Bern, Hinterkappelen, Switzerland; [10]Institute of Electronic Music and Acoustics, University of Music and Performing Arts Graz, Graz, Austria; [11]Department of Behavioral and Cognitive Biology, University of Vienna, Vienna, Austria; [12]Department of Evolutionary Biology, University of Vienna, Vienna, Austria; [13]Research Center on Animal Cognition, Center for Integrative Biology, CNRS - Paul Sabatier University, Toulouse, France; [14]Department of Integrative Biology, Oklahoma State University, Stillwater, United States; [15]Centro Jambatu de Investigación y Conservación de Anfibios, Quito, Ecuador; [16]Messerli Research Institute, University of Veterinary Medicine Vienna, Vienna, Austria

**\*For correspondence:**
andrius.pasukonis@gmc.vu.lt (AP);
loconnel@stanford.edu (LAO'C)

**Abstract** Sex differences in vertebrate spatial abilities are typically interpreted under the adaptive specialization hypothesis, which posits that male reproductive success is linked to larger home ranges and better navigational skills. The androgen spillover hypothesis counters that enhanced male spatial performance may be a byproduct of higher androgen levels. Animal groups that include species where females are expected to outperform males based on life-history traits are key for disentangling these hypotheses. We investigated the association between sex differences in reproductive strategies, spatial behavior, and androgen levels in three species of poison frogs. We tracked individuals in natural environments to show that contrasting parental sex roles shape sex differences in space use, where the sex performing parental duties shows wider-ranging movements. We then translocated frogs from their home areas to test their navigational performance and found that the caring sex outperformed the non-caring sex only in one out of three species. In addition, males across species displayed more explorative behavior than females and androgen levels correlated with explorative behavior and homing accuracy. Overall, we reveal that poison frog reproductive strategies shape movement patterns but not necessarily navigational performance. Together this work suggests that prevailing adaptive hypotheses provide an incomplete explanation of sex differences in spatial abilities.

## Editor's evaluation

In this important paper, the authors use intensive field monitoring and experimentally induced navigational challenges in three species of poison frog to examine two different hypotheses for sex differences in spatial ability. The study is simultaneously rich and complex; the results are solidly consistent with (but not necessarily definitively or exclusively in support of) the hypothesis that androgens may inadvertently affect spatial ability. This paper is of interest to organismal biologists and evolutionary scientists who study cognitive and behavioral sex differences including those with interests in the evolution of complex spatial behaviors.

## Introduction

Sex differences in spatial abilities are well established in mammals, where males tend to have larger home ranges and enhanced navigational skill compared to females (*Clint et al., 2012*; *Gray and Buffery, 1971*; *Jonasson, 2005*; *Jones et al., 2003*). In a series of comparative rodent studies, sex differences in space use have been linked to reproductive strategy, where polygamous species show sex differences in home range size and spatial abilities, but monogamous species do not (*Galea et al., 1994*; *Gaulin et al., 1990*; *Gaulin and FitzGerald, 1986*; *Gaulin and Fitzgerald, 1989*; *Sawrey et al., 1994*). In humans, men tend to score higher on spatial tests related to 3D mental rotations, whereas women tend to score better on object location memory (*Eals and Silverman, 1994*; *Silverman et al., 2007*; *Voyer et al., 1995*; reviewed in *Clint et al., 2012*; *Jones et al., 2003*). The adaptive specialization hypothesis has been proposed to explain sex differences in mammals and argues that enhanced spatial abilities in males is an adaptive trait, where males with better navigational skills and larger home ranges may have increased survival and reproductive success (*Gaulin and FitzGerald, 1986*; *Gaulin and Fitzgerald, 1989*; *Jones et al., 2003*). In addition, maternal care in mammals may limit space use and exploration in females (*Barnett and McEwan, 1973*; *Sherry and Hampson, 1997*; *Trivers, 1972*). While the adaptive specialization hypothesis has been widely accepted for decades, empirical support outside rodents is often inconclusive and alternative explanations are rarely evaluated.

The widely accepted adaptive explanations of sex differences were challenged by *Clint et al., 2012*. They countered that better spatial abilities in males might be a byproduct of sex differences in androgens unrelated to selective pressures on spatial behavior and reproductive strategies (i.e., the androgen spillover hypothesis). Cross-cultural studies show that human sex differences in spatial abilities are not universal, but are influenced by cultural and population-specific factors such as mobility related to lifestyle (*Berry, 1966*; *Cashdan et al., 2012*; *Jang et al., 2019*; *Trumble et al., 2016*). Higher androgen levels in mammals enhance spatial performance through effects on neural development and plasticity (*Dawson et al., 1975*; *Galea et al., 1995*; *Isgor and Sengelaub, 1998*; *Joseph et al., 1978*; *Neave et al., 1999*; *Roof and Havens, 1992*; *Schulz and Korz, 2010*; *Sherry and Hampson, 1997*; *Stewart et al., 1975*; *Van Goozen et al., 1995*; *Williams et al., 1990*). In humans, female performance in spatial ability tasks correlates positively with androgen levels and improves with androgen treatments (*Aleman et al., 2004*; *Burkitt et al., 2007*; *Driscoll et al., 2005*). From an adaptationist perspective, this relationship has been viewed as the proximate mechanism for the selection on males' spatial abilities. However, with few exceptions, empirical support for adaptive sex differences in spatial abilities is based on research in mammals, where males typically have higher androgen levels and larger home ranges than females (but see *Mabry et al., 2013*; *Mysterud et al., 2001*; *Ofstad et al., 2016*). To disentangle the effect of androgens and life-history traits on sex differences in spatial behavior, we need comparative research in groups of animals where females have more complex spatial behavior and are expected to have better spatial abilities than males.

Outside mammals, sex differences in spatial abilities have been studied in some birds and fishes, but the support for the adaptive specialization hypothesis remains equivocal. In brood-parasitic cowbirds, where females need to remember and locate multiple host nest sites, females have larger forebrain areas associated with spatial learning in comparison to males and non-parasitic relatives (*Sherry et al., 1993*; *Reboreda et al., 1996*). However, the outcomes of behavioral studies of sex difference in cowbirds' spatial abilities are inconclusive as some (*Guigueno et al., 2014*), but not others (*Astié et al., 1998*; *Astié et al., 2015*; *Lois-Milevicich et al., 2021*), found the expected female advantage in spatial tasks. In Blenniid fishes males defend small territories and perform parental care while females move between males, and have larger home ranges and larger forebrain areas associated with spatial

cognition than males (*Costa et al., 2011*). However, behavioral testing in one blenny species (*Fabre et al., 2014*) and some other fishes (*Keagy et al., 2019*; *Wallace and Hofmann, 2021*) found fewer or opposite sex differences than predicted by the adaptive specialization hypothesis. More taxonomically diverse study systems and explicit evaluation of alternative explanations are needed to test the adaptive specialization hypothesis and its broader implications for the evolution of vertebrate spatial cognition.

Amphibians show a remarkable variety of mating strategies and parental sex roles compared to mammals and birds, including widespread polyandry and male uniparental care (*Duellman, 1989*; *Schulte et al., 2020*; *Wells, 1977*; *Zamudio et al., 2016*). Such behavioral diversity provides natural comparison groups to test alternative hypotheses about sex differences in spatial abilities. In Neotropical poison frogs (Dendrobatoidea), male and female uniparental care, biparental care, and flexible parental roles occur among closely related species (*Schulte et al., 2020*; *Summers and Tumulty, 2014*; *Weygoldt, 1987*). Poison frog parental care involves complex spatial behavior, where parents navigate the rainforest to transport tadpoles from terrestrial clutches to pools of water (*Beck et al., 2017*; *Pašukonis et al., 2019*; *Ringler et al., 2013*). Poison frogs show well-developed spatial cognition and rely on spatial memory to relocate home territories and tadpole deposition sites (*Beck et al., 2017*; *Liu et al., 2016*; *Liu et al., 2019*; *Pašukonis et al., 2014*; *Pašukonis et al., 2016*; *Stynoski, 2009*). Male tadpole transport is the ancestral and most common form of parental care in poison frogs (*Carvajal-Castro et al., 2021*; *Summers and Tumulty, 2014*; *Weygoldt, 1987*), but female care and flexible parental roles have evolved in some species (*Fischer and O'Connell, 2020*; *Ringler et al., 2015b*). In species where females provide care, frogs place their tadpoles in small, resource-poor nurseries. Females frequently return to these pools to supplement the tadpoles' diet by provisioning with unfertilized eggs (*Brust, 1990*; *Summers and Tumulty, 2014*). All these parental behaviors require well-developed spatial memory and navigation abilities. Therefore, individuals with better spatial memory may have increased reproductive fitness, leading to enhanced spatial abilities in the sex that performs parental care, as proposed by the adaptive specialization hypothesis. Poison frogs also show sex-typical differences in androgen levels, where males have higher androgen levels than females, although androgen levels decrease during tadpole transport in males (*Fischer and O'Connell, 2020*). Thus, comparative studies in poison frogs provide a unique opportunity to understand how parental roles and reproductive strategies shape sex differences in space use and navigational abilities, and how hormones regulate these behaviors.

We conducted extensive field studies on sex differences in spatial behavior across three poison frog species with contrasting parental sex roles and reproductive strategies: the Brilliant-Thighed Poison Frog *Allobates femoralis*, an inconspicuous frog with flexible but predominantly male parental care, the Dyeing Poison Frog *Dendrobates tinctorius*, an aposematic frog with obligate male care, and the Diablito Poison Frog *Oophaga sylvatica*, an aposematic species with obligate female care. We tracked frogs with miniature tags to quantify sex differences in home range and parental care-associated space use in their natural environment. We then quantified the sex differences in navigational performance by experimentally translocating frogs from their home areas and tracking their homing behavior. We also used non-invasive methods to measure androgen levels before and after translocation. Based on the adaptive specialization hypothesis, we predicted that the tadpole transporting sex (males in *D. tinctorius* and *A. femoralis*, females in *O. sylvatica*) would have wider-ranging space use and better navigational performance. Following the androgen spillover hypothesis, we predicted that males would show enhanced navigation regardless of species differences in reproductive strategy.

## Results

### Sex differences in parental roles predict sex differences in space use

We first quantified the sex differences in space use and the association between movements and parental behavior across three species that differ in parental sex roles (*Figure 1*). In *A. femoralis*, where males do most parental care duties, the average male home range was 153% larger and movement extent 172% larger than females (*Figure 1*, *Table 1*, *Supplementary file 1*, *Table 1a*). We also found that long-term movements based on capture–recapture data were larger in males than in females of *A. femoralis* (LM: $\beta_{\text{extent male}}$ = 0.55, $t$ = 3.1, $n$ = 165, p = 0.002; *Figure 1—figure supplement*

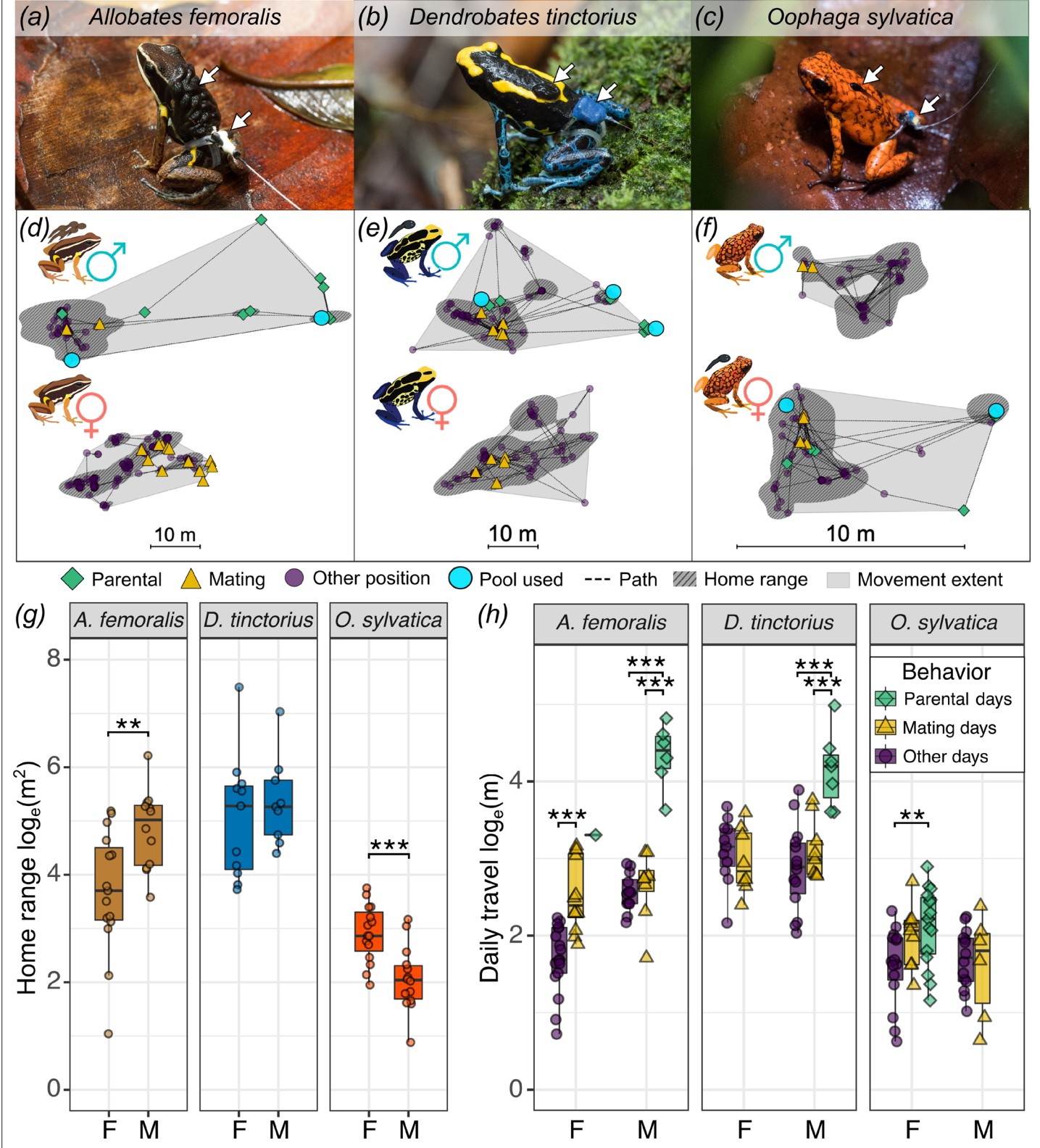

**Figure 1.** Parental sex roles and behavior drive sex differences in poison frog space use. Male (**a, b**) and female (**c**) individuals of each study species transporting tadpoles while wearing a tracking tag. White arrows indicate tadpoles and the tag. (**d, e**) Examples of representative space use patterns of one individual of each species and sex show different measured space use parameters. We calculated the daily travel as the cumulative distance (line) between all relocations (points) per day; the movement extent (gray shaded area); and the home range representing more intensely used areas (darker

*Figure 1 continued on next page*

*Figure 1 continued*

hatched area). Frog positions are classified to represent three types of behaviors associated with daily movements: parental behavior (green diamonds), mating behavior (yellow triangles), and other (purple circles). Light blue circles represent pools used for tadpole deposition. Note that the scale is different in the panel for (**f**) *O. sylvatica*. Boxplots show sex differences in home range size (**g**), and daily travel between days when parental behavior, mating, or neither were observed (**h**). Plot rectangles indicate the lower and upper quartiles with the median line, whiskers extend to 1.5 times the interquartile limited by value range, and dots indicate individuals. As frogs were tracked for multiple days, average values per individual per behavioral category are shown. Days were categorized as pool visits or mating days if the corresponding behavior was observed at least on one relocation of that day. *y*-Axes are $\log_e$-transformed. Statistical significance levels are indicated as **p < 0.01, ***p < 0.001.

The online version of this article includes the following figure supplement(s) for figure 1:

**Figure supplement 1.** Comparison of sex differences in movement extent between long-term recapture data and short-term tracking.

**Figure supplement 2.** Comparison of sex differences in daily travel and movement extent of *O.sylvatica* tracked in enclosures and a natural population.

1). In *D. tinctorius*, where males perform parental care, we found no significant sex differences in home range size or movement extent based on tracking data, but males showed wider-ranging long-term movements based on capture–recapture data (LM: $\beta_{\text{extent male}}$ = 0.43, $t$ = 2.62, $n$ = 154, p = 0.01; *Figure 1—figure supplement 1*). In *O. sylvatica*, where females perform parental care, the average male home range was 56% smaller and movement extent 57% smaller than in females. There was no sex difference in the *O. sylvatica* daily travel at the natural site (LMM: $\beta_{\text{daily male}}$ = −0.27, $t$ = −1.6, $n$ = 93, p = 0.1) nor within enclosures (LMM: $\beta_{\text{daily male}}$ = −0.11, $t$ = −1, $n$ = 287, p = 0.3), but short-term movement extent was larger in females than in males of *O. sylvatica* at both the natural site (LM: $\beta_{\text{extent male}}$ = −0.38, $t$ = −2.4, $n$ = 37, p = 0.02) and within enclosures (LM: $\beta_{\text{extent male}}$ = −0.42, $t$ = −2.9, $n$ = 29, p = 0.007; *Figure 1—figure supplement 2*).

Sex influenced the daily travel of *A. femoralis*, where the best-fit model included sex, behavior, daytime temperature, and random factors, but not in *D. tinctorius* and *O. sylvatica*. *A. femoralis* males moved more on days of parental care than on mating days (lsm contrast: $\beta$ = 1.8, p < 0.001) and other days (lsm contrast: $\beta$ = 1.9, p < 0.001), but equally between mating days and other days (lsm contrast p = 0.75; *Figure 1*). *A. femoralis* females moved more on days of mating than other days (lsm contrast: $\beta$ = 0.6, p < 0.001) and were only observed transporting tadpoles once (*Figure 1*). *D. tinctorius* males also moved more on days of parental care than on mating days (lsm contrast: $\beta$ = 1.3, p < 0.001) and other days (lsm contrast: $\beta$ = 1.5, p < 0.001), but moved equally between mating days and other days (lsm contrast p = 0.85, *Figure 1*). Daily travel of *D. tinctorius* females did not differ on days of mating from other days (lsm contrast p = 0.4, *Figure 1*), and females were never observed in the pools or with tadpoles. Daily travel of *O. sylvatica* males did not differ between mating days and other days (lsm contrast p = 0.95, *Figure 1*) and males were never seen transporting tadpoles, but were regularly found near the breeding pools often located inside their territories. Females of *O. sylvatica* moved more on days of parental care than on other days (lsm contrast: $\beta$ = 0.4, p = 0.004; *Figure 1*), but there was no difference between mating days and parental days (lsm contrast p = 0.8) or other days

**Table 1.** Home range size model summaries.

| Predictors | *A. femoralis* $\log_e$ (home range) | | *D. tinctorius* $\log_e$ (home range) | | *O. sylvatica* $\log_e$ (home range) | |
|---|---|---|---|---|---|---|
| | Estimates (CI) | p | Estimates (CI) | p | Estimates (CI) | p |
| (Intercept) | 3.15 (1.1 to 5.2) | **0.004** | 5.3 (4.0 to 6.6) | **<0.001** | 2.5 (1.6 to 3.4) | **<0.001** |
| Sex [male] | 0.9 (0.2 to 1.7) | **0.015** | 0.2 (−0.9 to 1.3) | 0.68 | −0.8 (−1.3 to 0.4) | **0.001** |
| Tracking duration | 0.06 (−0.1 to 0.2) | 0.46 | −0.02 (−0.1 to 0.06) | 0.62 | 0.04 (−0.04 to 0.1) | 0.36 |
| Observations | 25 | | 19 | | 29 | |
| $R^2$/$R^2$ adjusted | 0.28/0.21 | | 0.02/0.0 | | 0.40/0.35 | |

Summary of three linear models with $\log_e$-transformed home range size in *A. femoralis*, *D. tinctorius*, and *O. sylvatica* as the response variable, sex as the predictor, and tracking duration (in days) as a covariate. Statistical significance with p < 0.05 is highlighted in bold.

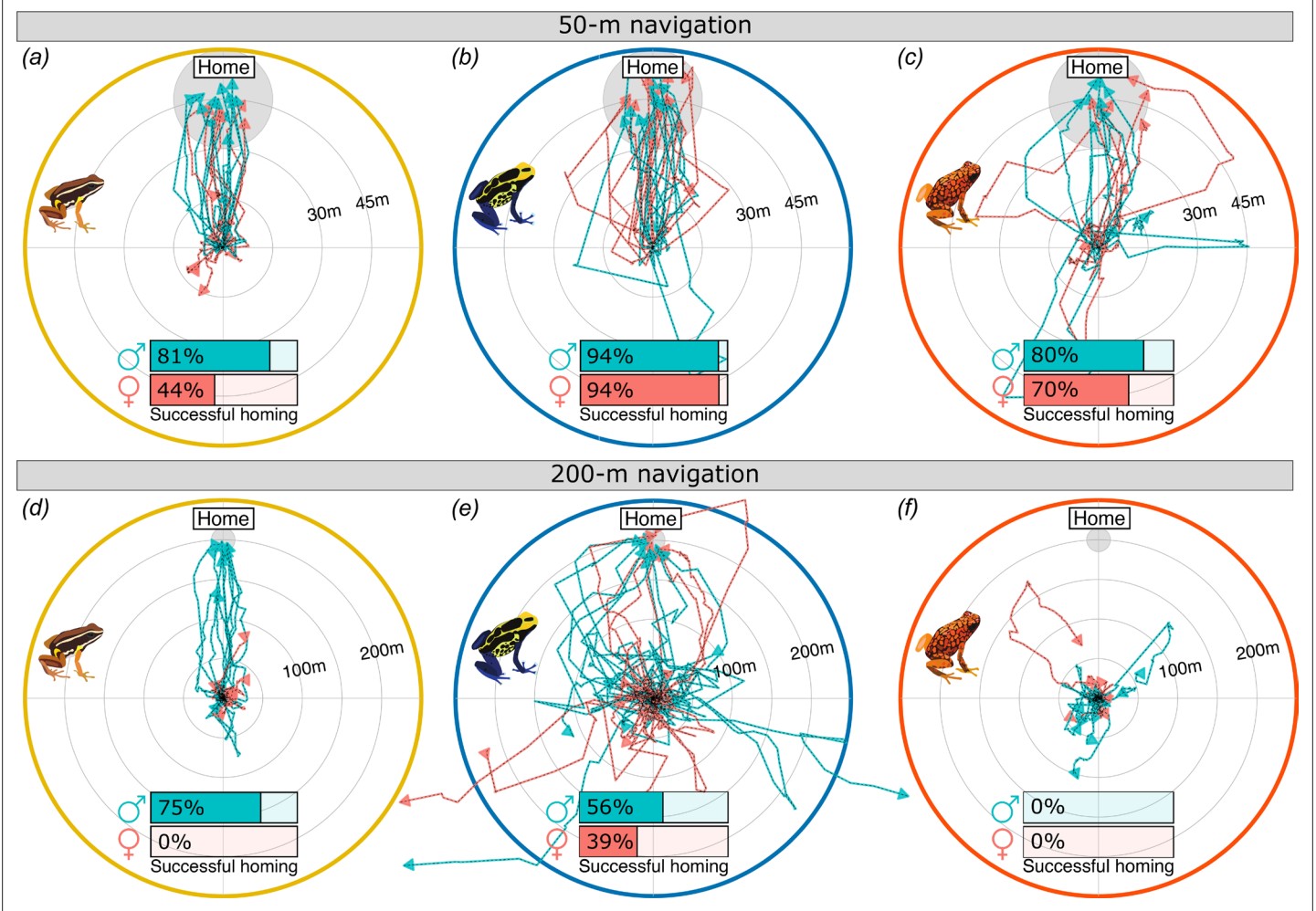

**Figure 2.** Species and sex differences in movement trajectories of translocated poison frogs. Homeward normalized movement trajectories of (**a**, **d**) *A. femoralis*, (**b**, **e**) *D. tinctorius*, and (**c**, **f**) *O. sylvatica* translocated approximately 50 m (**a–c**) or 200 m (**d–f**) from home. All trajectories are normalized to a common start location (center of the plot) and home direction (top of the plot). The approximate home area is indicated by a gray circle. Each line corresponds to a different individual with male trajectories in teal and female in red. The proportion of each sex that showed homing behavior is indicated on inserted bar plots. Frogs were considered homing if they completed at least 70% of the distance from the release site to the home center within 3 or 6 days for 50 and 200 m, respectively.

The online version of this article includes the following figure supplement(s) for figure 2:

**Figure supplement 1.** Sex differences in angular deviation from home direction.

**Figure supplement 2.** Temporal patterns of homing.

(lsm contrast p = 0.3). In summary, we found sex differences in space use that reflect sex differences in parental roles across species, and that parental care is associated with the longest movements in all three species.

## Sex differences in parental care do not predict navigational performance

We tested if there were sex differences in navigational performance that reflected sex differences in parental roles and space use across species. When translocated 50 m, *A. femoralis* males were more likely to return home (81% males, 44% females), but both sexes of *D. tinctorius* (94% males, 94% females) and *O. sylvatica* (80% males vs. 70% females) were equally likely to return (*Figure 2* and *Table 2*). When translocated 200 m, only males of *A. femoralis* (75% males, 0% females) and both females and males of *D. tinctorius* (56% males, 39% females), but no *O. sylvatica* were able to return home (*Figure 2* and *Table 2*). One *O. sylvatica* male could not be located after 6 days and was found

**Table 2.** Homing success model summaries.

| Predictors | *A. femoralis* 50 m homing success | | *D. tinctorius* 200 m homing success | | *O. sylvatica* 50 m homing success | |
|---|---|---|---|---|---|---|
| | Log-Odds (CI) | p | Log-Odds (CI) | p | Log-Odds (CI) | p |
| (Intercept) | 107.8 (34.7 to 242.3) | **0.02** | 18.7 (−23.9 to 65.4) | 0.4 | 152.65 (−127.6 to 543.3) | 0.35 |
| Sex [male] | 5.2 (1.7 to 12.4) | **0.03** | 1.05 (−1.2 to 3.6) | 0.38 | 1.4 (−1.0 to 4.6) | 0.29 |
| Temp. | −4.8 (−10.8 to 1.8) | **0.020** | −0.8 (−2.7 to 0.85) | 0.35 | −6.5 (−23.3 to 5.5) | 0.35 |
| Weight | 4.7 (−1.9 to 13.0) | 0.19 | 0.2 (−1.0 to 1.5) | 0.74 | NA | |
| Observations | 32 | | 32 | | 18 | |
| $R^2$ Tjur | 0.59 | | 0.07 | | 0.10 | |

Summary of three logistic regression models with homing success in *A. femoralis*, *D. tinctorius*, and *O. sylvatica* as the response variable, sex as the predictor, and average daytime temperature (Temp.) and frog weight as covariates. We did not perform statistical comparisons conditions *A. femoralis* 200 m because only males successfully returned; for *O. sylvatica* 200 m because no frogs returned and for *D. tinctorius* 50 m because both sexes return at an equal rate. Weight was excluded for *O. sylvatica* to achieve model convergence. Statistical significance with p < 0.05 is highlighted in bold.

back home 2 days later. *A. femoralis* were less likely to return home with higher daytime temperature, but the daytime temperature had no effect on homing success in *D. tinctorius* and *O. sylvatica* (*Table 2*). Frog weight had no effect on homing success. When translocated 50 m, *A. femoralis* males explored larger areas than females, but we observed no sex difference in *D. tinctorius* and *O. sylvatica* (*Figure 3*, *Supplementary file 1A*, *Table 1b*). When translocated 200 m, males of all three species were more explorative than females (*Figure 3* and *Table 3*). In addition, *A. femoralis* explored less with higher daytime temperature, but the daytime temperature did not affect the explored area in *D. tinctorius* or *O. sylvatica* (*Table 3*, *Supplementary file 1*, *Table 1b*). Frog weight positively influenced the explored area only in *D. tinctorius* translocated 200 m (*Table 3*).

After 50-m translocation, *A. femoralis* males returned more directly and faster than females (*Figure 3*, *Table 4* and *Supplementary file 1*, *Table 1c*). After moving ~20% of the translocation distance from the release site, *A. femoralis* males but not females were significantly oriented toward home (50 m: male Rayleigh p < 0.001, female Rayleigh p = 0.15; 200 m: male Rayleigh p < 0.001; *Figure 2—figure supplement 1*). We also found a sex difference in the distribution of angular deviations in *A. femoralis* (F(22) = 3.9, p = 0.03). In *D. tinctorius*, females returned in more direct paths than males from 200 m (*Figure 3* and *Table 4*). Both males and females of *D. tinctorius* were significantly oriented when translocated 50 m (male Rayleigh p = 0.007, female Rayleigh p < 0.001; *Figure 2—figure supplement 1*), but not when translocated 200 m (male Rayleigh p = 0.4, female Rayleigh p = 0.85; *Figure 2—figure supplement 1*). There were no sex differences in the distribution of angular deviations in *D. tinctorius*. In *O. sylvatica*, we found no sex difference in trajectory straightness, homing duration, or the distribution of angular deviations. After moving ~20% of the translocation distance, males but not females of *O. sylvatica* were significantly oriented when translocated 50 m (male Rayleigh p = 0.015, female Rayleigh p = 0.18; *Figure 2—figure supplement 1*), but not after 200 m translocation (male Rayleigh p = 0.9, female Rayleigh p = 0.9; *Figure 2—figure supplement 1*). *A. femoralis* and *O. sylvatica* returned in less direct paths and slower with higher daytime temperature. *Dendrobates tinctorius* returned slower with higher daytime temperature, but temperature had no effect on trajectory straightness (*Table 4*, *Supplementary file 1*, *Table 1c*). Frog weight did not affect the homing duration.

To sum up our navigation experiments, we found that *A. femoralis* males navigate home faster and more accurately than females, *D. tinctorius* females navigate more accurately than males from long distances, and males across species display more exploration-related movement compared to females regardless of sex differences in parental care roles.

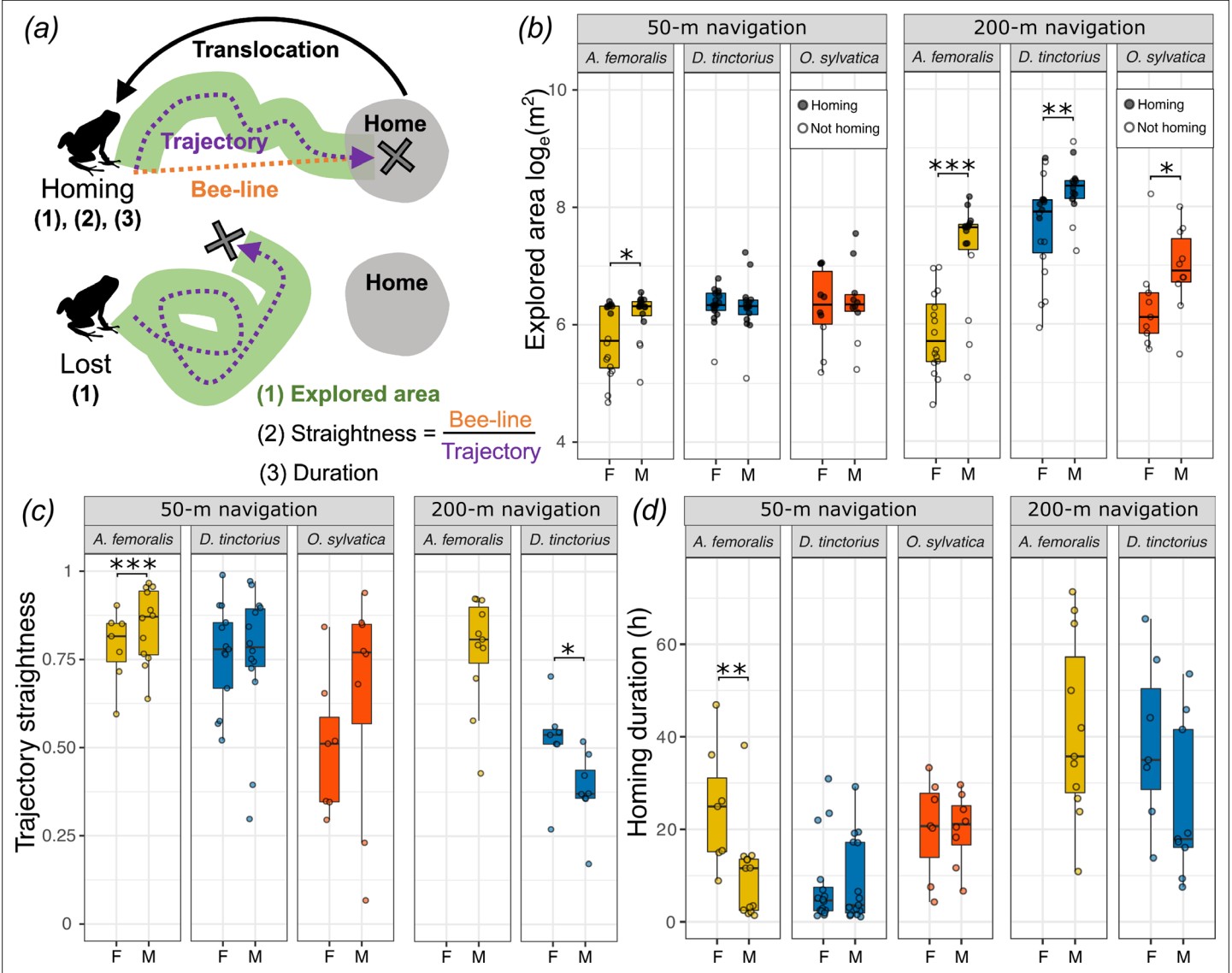

**Figure 3.** Sex differences in poison frog exploration and navigational performance. (**a**) Schematic representation of the parameters measured during navigation experiments (numbers in parentheses), which are plotted in panels (**b–d**). Explored area (**b**), trajectory straightness (**c**), and homing duration (**d**) were measured for successful homing, while only explored area (**b**) were measured for frogs that did not return home. Boxplots show sex differences in (**b**) explored area (log_e-transformed), (**b**) homing trajectory straightness, and (**c**) homing duration (**a**). Filled and empty circles indicate individuals that were homing or not. Plot rectangles indicate the lower and upper quartiles with the median line, whiskers extend to 1.5 times the interquartile limited by the value range, and dots indicate individuals. Statistical significance levels are indicated as *p 0.05–0.01, **p < 0.01, ***p < 0.001.

## Androgens correlate with navigation-associated behavior

We also investigated the relationship between androgen levels and spatial behavior during the navigation task described above. There was high interindividual variation of androgen levels in both sexes of all species, but on average, males showed higher androgen levels in all three species (*Figure 4*, *Supplementary file 1A*, *Table 1d*). There were no significant differences between baseline and back-home samples in all three species. Baseline levels did not influence exploration, homing duration, nor trajectory straightness in *A. femoralis* (*Figure 4*, *Supplementary file 1A*, *Table 1e*). Baseline androgen levels, together with sex, translocation distance, and frog weight significantly predicted exploration in *D. tinctorius*, but did not influence homing duration or trajectory straightness (*Figure 4*, *Supplementary file 1A*, *Table 1f*). Baseline levels significantly predicted trajectory straightness, but not exploration or homing duration in *O. sylvatica* (*Figure 4*, *Supplementary file 1*, *Table 1g*). The explored area

**Table 3.** Explored area model summaries.

| Predictors | A. femoralis 200 m log$_e$ (explored area) | | D. tinctorius 200 m log$_e$ (explored area) | | O. sylvatica 200 m log$_e$ (explored area) | |
|---|---|---|---|---|---|---|
| | Estimates (CI) | p | Estimates (CI) | p | Estimates (CI) | p |
| (Intercept) | 28.0 (8.7 to 47.4) | **0.006** | 5.6 (−7.65 to 18.9) | 0.39 | 30.85 (−97.0 to 158.7) | 0.61 |
| Sex [male] | 1.5 (0.9 to 2.1) | **<0.001** | 1.5 (0.8 to 2.2) | **<0.001** | 0.8 (0.02 to 1.7) | **0.045** |
| Temp. | −1.0 (−1.8 to 0.2) | **0.017** | −0.04 (−0.6 to 0.5) | 0.87 | −0.9 (−6.4 to 4.5) | 0.72 |
| Weight | 1.0 (−0.5 to 2.5) | 0.19 | 0.6 (0.2 to 0.9) | **0.004** | −1.4 (−3.2 to 0.3) | 0.10 |
| Observations | 30 | | 32 | | 15 | |
| $R^2$/$R^2$ adjusted | 0.56/0.51 | | 0.41/0.35 | | 0.38/0.21 | |

Summary of three linear models with log$_e$-transformed explored area in *A. femoralis*, *D. tinctorius*, and *O. sylvatica* as the response variable, sex as the predictor, and average daytime temperature (Temp.) and frog weight as covariates. Statistical significance with p < 0.05 is highlighted in bold.

had a significant positive effect and successful homing a significant negative effect on delta androgen levels in *A. femoralis*, but not in *D. tinctorius* and *O. sylvatica* (**Supplementary file 1**, **Table 1h**).

## Discussion

Sex differences in spatial behaviors are typically interpreted through the lens of the adaptive specialization hypothesis, where larger home ranges and better navigational abilities in males are seen as adaptive traits (**Jones et al., 2003**). This has been countered with the androgen spillover hypothesis, which suggests that enhanced spatial abilities in males may be a byproduct of higher male androgen levels rather than an adaptation (**Clint et al., 2012**). However, there are no comparative studies where females and males of closely related species have reversed spatial behavior and are expected to show a reversal in spatial abilities. Here, we linked the reproductive strategies, space use, navigational performance, and androgen levels in three species of frogs that differ in which sex performs spatially relevant parental care tasks that tie spatial accuracy to reproductive fitness (**Figure 5**). We found that parental care shapes sex differences in space use, but no evidence that sex differences in

**Table 4.** Homing trajectory straightness model summaries.

| Predictors | A. femoralis 50 m homing straightness | | D. tinctorius 50 m homing straightness | | D. tinctorius 200 m homing straightness | | O. sylvatica 50 m homing straightness | |
|---|---|---|---|---|---|---|---|---|
| | Estimates | p | Estimates | p | Estimates | p | Estimates | p |
| (Intercept) | 28.7 (12.6 to 44.9) | **<0.001** | 2.0 (−18.9 to 22.9) | 0.85 | 8.9 (−0.7 to 18.5) | 0.069 | 111.0 (25.9 to 196.0) | **0.011** |
| Sex [male] | 0.7 (0.1 to 1.3) | **0.017** | −0.05 (−0.8 to 0.7) | 0.89 | −0.5 (−0.9 to 0.02) | **0.039** | 0.5 (−0.4 to 1.3) | 0.30 |
| Temp. | −1.1 (−1.8 to 0.5) | **0.001** | −0.04 (−0.9 to 0.8) | 0.93 | −0.4 (−0.75 to 0.03) | 0.071 | −4.8 (−8.4 to 1.1) | **0.011** |
| Observations | 17 | | 22 | | 15 | | 13 | |
| $R^2$ | 0.35 | | 0.001 | | 0.39 | | 0.44 | |

Summary of four beta regression models with homing trajectory straightness in *A. femoralis*, *D. tinctorius*, and *O. sylvatica* as the response variable, sex as the predictor, and average daytime temperature (Temp.) as a covariates. Statistical significance with p < 0.05 is highlighted in bold.

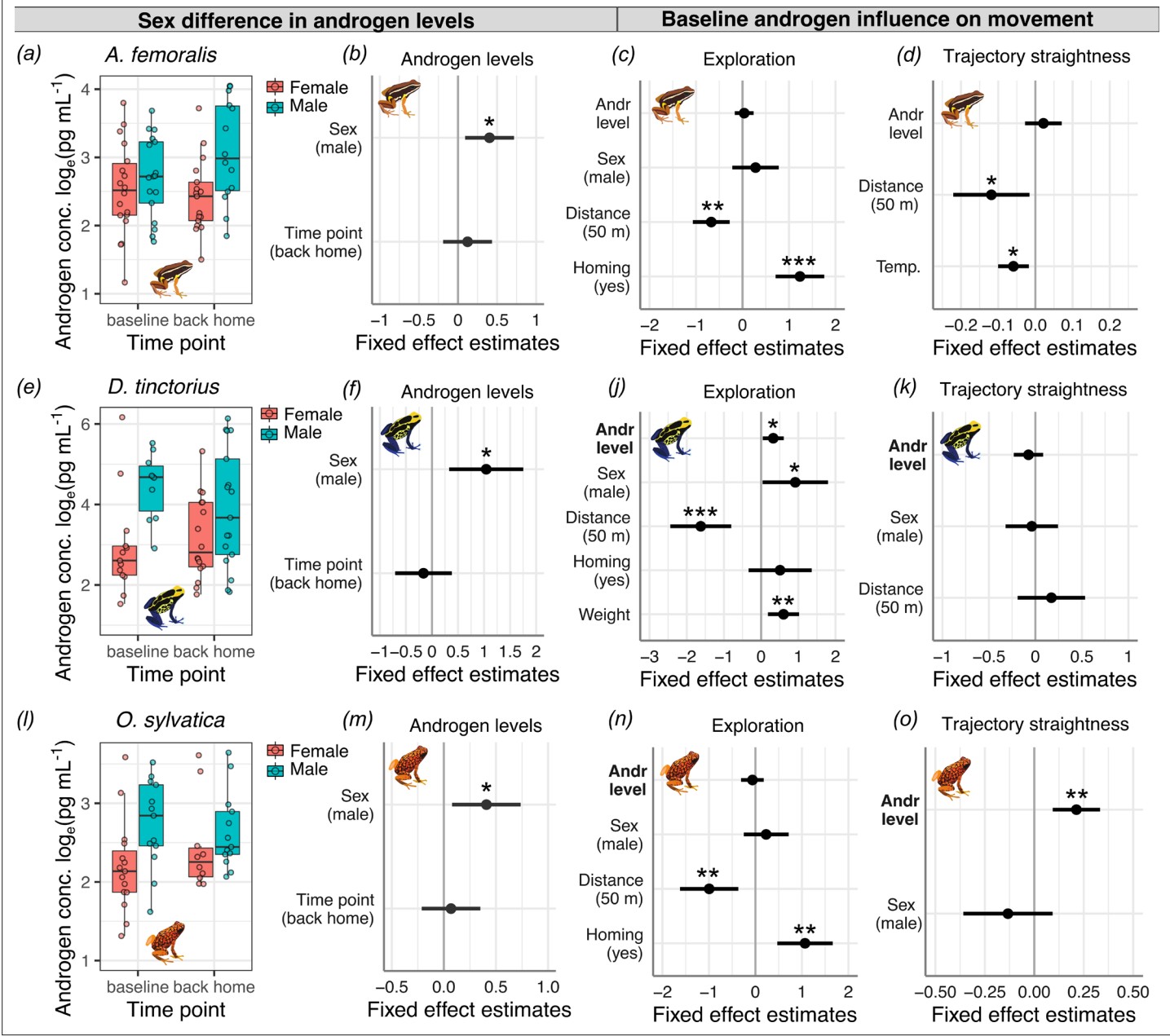

**Figure 4.** Relationships between androgen levels and spatial behavior. Boxplots show sex differences in water-borne androgen concentration measured before and after the navigational task in (**a**) *A. femoralis*, (**e**) *D. tinctorius*, and (**l**) *O. sylvatica*. The coefficient plots indicate the effect size and confidence intervals of androgen level difference between sexes and the two sampling points for (**b**) *A. femoralis*, (**f**) *D. tinctorius*, and (**m**) *O. sylvatica* and the influence of androgen levels and other factors on exploration (**c, j, n**) and homing trajectory straightness (**d, k, o**) in each species. The plot title represents the response variable of the respective regression model with its predictors on the *y*-axis, and fixed effect estimates (black dots) ± 95% confidence interval (error bars) on the *x*-axis. For categorical predictors such as 'sex', the estimates are always shown for the reference factor (its label in parenthesis), which is compared to the other factors represented by the intercept. The overlap of the error bars with the zero-reference line indicates lack of a significant effect on the response variable. All continuous predictors were centered and standardized. Androgen concentrations are natural $\log_e$-transformed. Statistically significant levels are indicated as *p 0.05–0.01, **p < 0.01, ***p < 0.001.

navigational performance are linked to the reproductive strategy. Importantly, we found that females did not outperform males in *O. sylvatica*, the species with more complex female spatial behavior and larger home ranges associated with female parental care. We also found that males of all three species tended to be more explorative than females and had higher androgen levels. Moreover, increased androgen levels were associated with higher exploration in two species with male care and

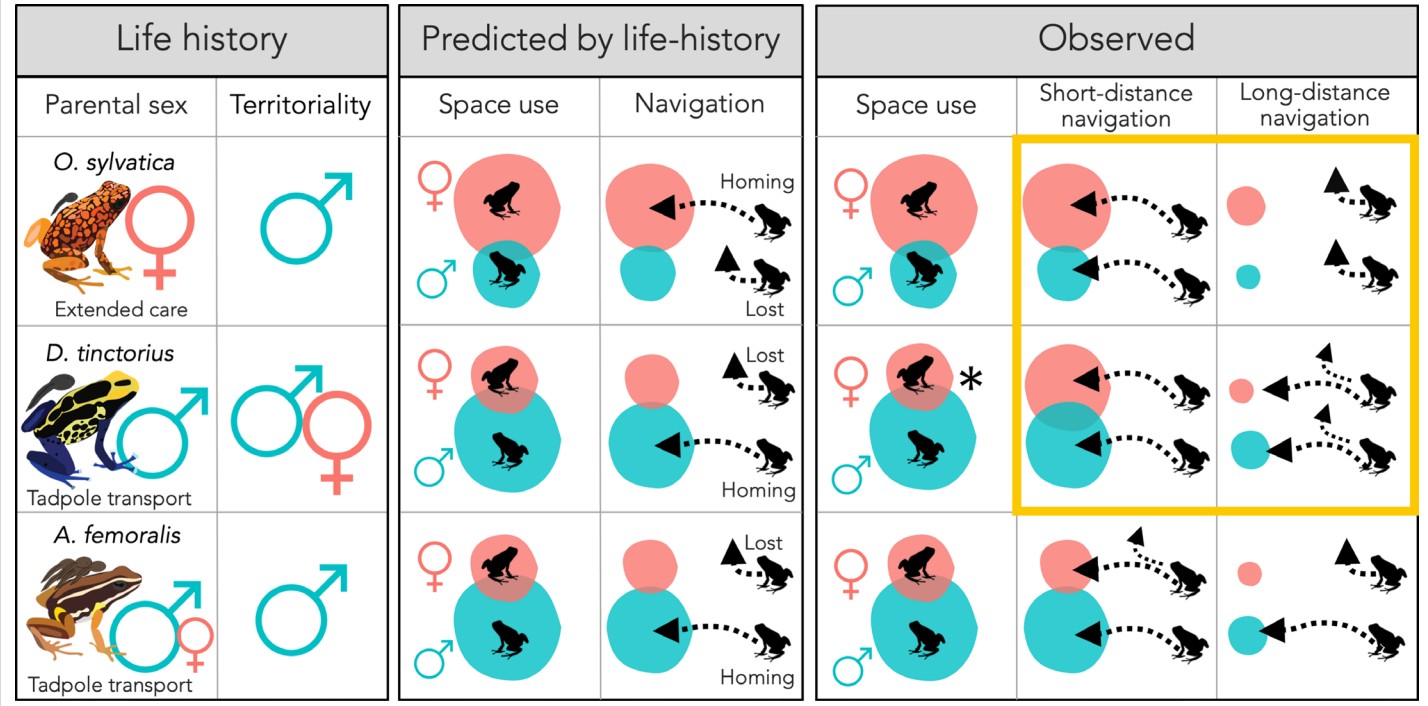

**Figure 5.** Summary of sex and species differences in life-history traits, predicted, and observed space use and navigational performance. Pink and teal circles represent the space use extent for females and males, respectively, and dotted arrows represent homing success after translocations. Arrows pointing to home range represent successful homing and arrows pointing away present lost frogs, based on the overall pattern of homing success. Note that possible differences in homing accuracy and duration are not represented. Yellow frame highlights the observed outcomes that differed from the predictions and in all of these cases we found no sex differences where differences were expected. *Sex difference only observed in long-term movements and qualitatively in vertical space use.

navigational accuracy in the species with female care, leaving open the possibility that sex differences in spatial behavior might result from sex differences in androgen levels independent of the differences in parental sex roles.

## Reproductive strategy shapes species and sex differences in space use

We show that parental care in poison frogs increases the mobility of the caring sex and thus shapes the sex differences in space use. In *D. tinctorius* and *A. femoralis*, males transport tadpoles outside their territory and provide no further care. As we predicted, *A. femoralis* males have a much wider space use than females. Male long-distance movements were primarily observed during tadpole transport and occasional territory shifts, while female space use extent was mainly driven by mate-seeking. This sex difference was less pronounced in *D. tinctorius*, where males did not differ from females in short-term space use based on tracking. Males, however, moved over a wider range based on long-term capture–recapture data. The apparent difference between short-term tracking and long-term recapture data in *D. tinctorius* could be due to real sex differences only emerging in longer-term space use patterns, or the limited statistical power to detect a slight sex difference with a much smaller sample size in the tracking dataset. *D. tinctorius* males also moved the longest distances on tadpole transport days. During tadpole transport, males often climbed vertically to reach water-filled tree holes high above the ground, which they are known to use for tadpole deposition (*Fouilloux et al., 2021*). Our space-use data do not capture these vertical movements and thus might considerably underestimate *D. tinctorius* male mobility and associated sex differences in vertical space use qualitatively and quantitatively. We could not identify what drove wide-ranging horizontal movements in *D. tinctorius* females, but seeking better foraging sites has been suggested among the movement drivers in *D. tinctorius* (*Born et al., 2010*).

Including a species with female uniparental care whose spatial movements are tied to reproductive fitness allowed us to ask whether enhanced mobility could be an adaptive trait linked to parental care.

*O. sylvatica* females must remember and revisit tadpole nurseries dispersed over tens of meters to provision their young with trophic, unfertilized eggs (*Stynoski, 2009*; *Summers, 1992*; this study). As predicted, tracking revealed larger female home ranges in *O. sylvatica.* Females typically moved between several reproductive pools and moved most when visiting or switching between the pool sites. Like *D. tinctorius* males, *O. sylvatica* females regularly climbed vertically to water-filled plants up to 10 m above the ground, but these vertical movements are not captured in our space-use data. Therefore, our data underestimated the mobility of *O. sylvatica* females and the resulting magnitude of the sex difference. In contrast, males did not climb above 2 m, and their movements were restricted to exceedingly small calling territories. It remains unknown if males explore larger areas over the long term, particularly when searching for new territories. Overall, *O. sylvatica* showed more restricted movements than the two species with predominantly male parental care. Several previous studies in poison frogs indicate that species with extended female care, which includes tadpole provisioning, show more restricted space use than closely related species with male uniparental care (*Brown et al., 2009*; *Donnelly, 1989*; *McVey et al., 1981*; *Murasaki, 2010*; *Pašukonis et al., 2019*; *Summers, 1992*). Our data confirm that sex and species differences in space use across poison frogs can be largely explained by species differences in the care-providing sex and the intensity of parental care.

Research on sex differences in space use has typically focused on differences in mating strategies. For example, male bias in larger home ranges is well documented in rodents, where polygamous species tend to have greater sex differences than monogamous species (reviewed in *Clint et al., 2012*; *Jones et al., 2003*). In lizards, a meta-analysis of home range size in 60 species found that males have larger home ranges than females, and suggested that this sex difference is related to the mating system and access to females (*Perry and Garland, 2002*). Our study shows that parental care can directly influence the space use of the caring sex resulting in sex differences in space use. Parental care has been hypothesized to reduce mobility in the caring sex because the research has focused on maternal care in lactating mammals (*Barnett and McEwan, 1973*; *Sherry and Hampson, 1997*; *Trivers, 1972*). In contrast, moving with the offspring or for offspring provisioning is taxonomically widespread in vertebrates and invertebrates (*Choe and Crespi, 1997*; *Clutton-Brock, 1991*; *Royle et al., 2012*) and can increase mobility in the caring sex, thereby shaping sex differences in space use patterns in taxonomically diverse groups.

Understanding sex differences in spatial behavior requires sampling at different spatial and temporal scales. The home range (where the animal spends most of the time) and the full movement extent (which includes less frequent excursions) showed the same pattern of sex and species differences. However, we observed fewer sex differences in the daily travel distances than in measures of the area extent used. For example, *O. sylvatica* males and females travel similar daily distances, but male movement is restricted to small territories. Females, in contrast, move over larger areas resulting in larger home ranges, which are used less intensely than male territories. Higher mobility in a restricted area, such as seen in territorial males of *O. sylvatica*, might implicate more fine-scale spatial knowledge, which our study does not test. On the other hand, a larger extent of the area used may implicate better-developed long-distance navigational skills, which were the target of our study. Sampling method and period also have a strong influence on the measured space use patterns. For example, sex difference in movement extent of *D. tinctorius* only became apparent when looking at the long-term capture–recapture data. Obtaining long-term movement patterns under natural conditions is among the top future challenges to fully appreciate the sex- and species-specific selective pressures on animal movement and spatial abilities.

## Species differ in navigational and movement strategy

All three species showed well-developed homing ability, which is consistent with previous studies in amphibians, including various anurans (*Arcila-Pérez et al., 2020*; *Dole, 1968*; *McVey et al., 1981*; *Navarro-Salcedo et al., 2021*; *Navarro-Salcedo et al., 2022*; *Pašukonis et al., 2014*; *Pašukonis et al., 2018*; *Pichler et al., 2017*; *Shaykevich et al., 2021*; *Sinsch, 1987*; *Sinsch, 1992*) and caudates (*Diego-Rasilla et al., 2005*; *Grant et al., 1968*; *Joly and Miaud, 1989*; *Phillips et al., 1995*; *Sinsch, 2007*; *Twitty et al., 1964*; reviewed in *Ferguson, 1971*; *Sinsch, 2006*; *Wells, 2010*). Despite limited movement capacity and sedentary lifestyle, many amphibians appear to share a general ability to navigate home after translocations from distances exceeding their routine movements (*Sinsch, 1990*; *Sinsch, 2014*). Moreover, the fact that species showing such tremendous variation in life history share

this capacity suggests that well-developed navigational abilities play a fundamental role in amphibian reproduction and survival. Like many other tropical anurans, poison frogs rely on moving between small and scattered water bodies for reproduction, which might have selected for particularly highly developed navigational abilities in this group.

The scale and strategy of navigation varied between species and were related to species differences in home range size and reproductive strategy. For example, *O. sylvatica* did not return from longer translocation distances, in line with their much smaller home ranges relative to *A. femoralis* and *D. tinctorius*. This apparent correlation between home range size and homing performance further supports the hypothesis that poison frogs rely on exploration and spatial learning for navigation (*Pašukonis et al., 2014*; *Pašukonis et al., 2016*; *Pašukonis et al., 2018*). Surprisingly, the distances from which poison frogs navigate back home exceed the extent of observed routine movements (*Pašukonis et al., 2014*; *Pašukonis et al., 2018*; this study) leaving many open questions about the possible cues learned and used for orientation. Since we did not scale the translocation distances in the navigation task to the home range size of each species, species with smaller home range, such as *O. sylvatica,* were translocated much longer distance in relation to their routine movements. Given the much smaller movement extent of *O. sylvatica* compared to the other two species, their behavior after the short distance translocation might be better compared to the long-distance translocation in the other two species.

While species differences must be interpreted with caution, our results indicate that navigational and movement strategies can differ strikingly even between closely related species (*Figure 2*, *Figure 2—figure supplement 2*). After translocation, *A. femoralis* tended to stay close to the release site for prolonged periods and then navigate back home via a direct path. Non-homing individuals moved very little and remained close to the release site. However, *D. tinctorius* showed wide-ranging exploratory movements and usually returned home through an indirect and lengthy route. Similarly, even though *O. sylvatica* returned home only from shorter distances, they also showed some wide-ranging exploratory behavior that was never observed in *A. femoralis*. We hypothesize that some of these species differences could be linked to movement strategy differences selected under different predation pressure. *D. tinctorius* and *O. sylvatica* are brightly colored species that forage actively to acquire their alkaloid-based chemical defenses from the invertebrate diet (*Santos et al., 2016*; *Santos and Cannatella, 2011*). Aposematic coloration may reduce predation pressure and the cost of movement while potentially increasing exploration (*Carvajal-Castro et al., 2021*; *Pough and Taigen, 1990*; *Speed et al., 2010*; *Summers, 2019*; *Toft, 1981*), allowing different navigational strategies. *A. femoralis*, on the other hand, is cryptically colored, non-toxic, and a generalist sit-and-wait forager. The only two predation events observed during tracking occurred in *A. femoralis* and we have regularly observed predation and predation-related injuries during the long-term capture–recapture monitoring in *A. femoralis*, but not in *D. tinctorius*. Thus, the predation pressure is likely to be much higher and the movement more costly for cryptic species, potentially selecting more efficient orientation strategies. These differences between closely related species provide a remarkable system for future work on the selective pressures shaping the animal movement and navigational strategy.

## Navigational performance provides no evidence for adaptive sex differences

The adaptive specialization hypothesis is the leading hypothesis to explain variation in home range size and spatial memory between species and sexes in mammals. It predicts that adaptive sex differences in navigational ability are linked to life-history traits. We found no evidence for adaptive sex differences in navigational ability in poison frogs. Contrary to our prediction, we found no sex difference in the navigational performance of *D. tinctorius*, a species with male uniparental care. Not only did males not outperform females, but females even showed slightly more accurate homing trajectories than males when navigating long distances. However, the lack of sex differences in the navigational performance of *D. tinctorius* somewhat fits the limited sex differences in space use observed in this species. Crucially, although females of *O. sylvatica* have larger space use and perform tadpole transport and egg provisioning, we found no sex differences in the navigational performance. Previous research on amphibians has also shown patterns inconsistent with sex differences in spatial abilities being an adaptive trait linked to reproductive strategy. Place discrimination tasks have not revealed consistent sex differences in *Engystomops pustulosos*, a frog species where females seek for and move between

males (*Liu and Burmeister, 2017*; *Ventura et al., 2019*). Using translocation and recapture methods, one recent study in *Andinobates bombetes*, a poison frog with male uniparental care, found no sex differences in homing rates after translocation (*Arcila-Pérez et al., 2020*). Another recent recapture study in the glass frog *Ikakogi tayrona*, a species with prolonged maternal care and male territoriality, found that only males, but not females, showed homing after translocations (*Navarro-Salcedo et al., 2022*). Together, our findings and the growing literature on amphibian navigation parallel findings in mammals, where males typically outperform females in spatial tasks or no sex differences are found.

The only species with marked sex differences in navigational performance was *A. femoralis*, where males were more likely to return home, returned from longer distances, and returned faster and more accurately than females. This finding is partially unexpected because *A. femoralis* females commute between males for reproduction (*Fischer et al., 2020*; *Ringler et al., 2012*) and remember the exact locations of their clutches (*Ringler et al., 2016b*). However, our task did not test for place learning but for large-scale navigation and our space-use data clearly show that males move over a larger extent than females in both the short and long term. As non-homing *A. femoralis* females typically stay stationary, it is difficult to disentangle the lack of motivation from their inability to return home. We expected that *A. femoralis* females would be motivated to return home because they show site fidelity (*Fischer et al., 2020*; *Ringler et al., 2009*; this study) and monitor the presence of their mating partners to eventually take over tadpole transport in case of male absence (*Ringler et al., 2015b*). Indeed, females returned, albeit slower from shorter translocation distances, indicating that they were motivated to return in a sufficiently familiar area. In addition, homing females showed less directed homing, suggesting a sex difference in orientation accuracy. However, males may be more motivated to return home quickly because they risk losing both their territory and all current offspring due to territorial takeovers and potential cannibalism by other males (*Ringler et al., 2017*). We believe that male *A. femoralis* likely have better navigational abilities than females, but the motivational state linked to each individual's current reproductive or parental status may explain some of the sex and interindividual differences observed in homing performance.

Even though our data do not fit the generic predictions of the adaptive specialization hypothesis, it also do not rule it out. Interestingly, we found sex differences only in the non-toxic *A. femoralis*, which likely faces much higher predation pressure than the two aposematic species. It is possible that adaptive sex differences in navigational ability emerge only under strong predation pressure, explaining the lack of selective pressure and sex difference in aposematic *O. sylvatica* and *D. tinctorius*. Furthermore, our spatial task specifically tested for large-scale navigation. While there is clear evidence that poison frogs rely on learning for spatial orientation (*Beck et al., 2017*; *Liu et al., 2016*; *Liu et al., 2019*; *Pašukonis et al., 2014*; *Pašukonis et al., 2016*; *Pašukonis et al., 2019*), the exact cognitive mechanisms remain unknown. It is possible that smaller scale spatial tasks specifically testing for place learning and recall (e.g., *Liu et al., 2019*) would reveal sex differences in specific spatiocognitive abilities, which is particularly relevant for locating clutches and breeding pool in poison frogs (*Beck et al., 2017*; *Pašukonis et al., 2014*; *Ringler et al., 2016a*; *Stynoski, 2009*). In human studies, naturalistic navigation tasks, such as pointing to a distant location, reveal fewer sex differences (*Cashdan et al., 2012*; *Jang et al., 2019*; *Trumble et al., 2016*) than tasks testing specific spatiocognitive ability, such as three-dimensional rotation or object location memory (*Silverman et al., 2007*; *Voyer et al., 1995*). Sex differences are also task specific in cowbirds (*Astié et al., 1998*; *Astié et al., 2015*; *Lois-Milevicich et al., 2021*) and cichlid fish (*Wallace and Hofmann, 2021*). Therefore, understanding the specific ecological context of each species is key to identify the spatiocognitive abilities potentially under selection and formulating testable hypotheses. Naturalistic experiments targeting several cognitive abilities in poison frogs and other animals will be crucial to test the adaptive specialization and other alternative explanations.

## Males explore more than females

Males are bolder and more explorative in several species and taxa (e.g., fish: *Harris et al., 2010*; *King et al., 2013*; bird: *Schuett and Dall, 2009*) and multiple adaptive hypotheses have been proposed to explain increased exploration (reviewed in *Schuett et al., 2010*; *Trivers, 1972*). Sex differences in exploration tendency could be connected to the sex-biased dispersal observed in different mating strategies, where male-biased dispersal is common in polygamous mammals while female-biased dispersal is common in monogamous birds (*Greenwood, 1980*; *Li and Kokko, 2019*; *Mabry et al.,*

*2013*). However, in birds and mammals, sex, mating systems, and parental care are tightly linked, making it difficult to disentangle factors shaping sex differences in exploratory behavior. In the present study, males of all three species tended to be more explorative than females, particularly when translocated longer distances and, therefore, in less familiar environments. Even in *O. sylvatica*, a species where females perform parental care and have wider space use, males tended to be more explorative after translocations. Male-biased dispersal and higher male exploration rates have also been observed in some frogs without parental care (*Engystomops pustulosus*: *Lampert et al., 2003*; *Bufo bufo*: *Ogurtsov et al., 2018*; *Xenopus tropicalis*: *Videlier et al., 2015*). Thus, regardless of parental care strategies, different life histories, and sex differences in home range size, male amphibians tend to be more exploratory, suggesting that other factors, such as male-biased dispersal and high intrasexual male competition may be associated with the sex difference in exploration.

## Linking androgens to exploration and navigation

Androgens have been linked to spatial abilities in mammals for several decades (*Dawson et al., 1975*; *Galea et al., 1995*; *Isgor and Sengelaub, 1998*; *Joseph et al., 1978*; *Schulz and Korz, 2010*; *Sherry and Hampson, 1997*; *Stewart et al., 1975*; *Williams et al., 1990*). *Clint et al., 2012* proposed that the often-observed male superiority in spatial navigation might be a side effect of sex difference in androgen levels rather than an adaptation to direct selective pressures on males' spatial abilities. Our results are somewhat in line with this hypothesis as females did not outperform males when expected based on adaptive predictions. Additionally, although we did not observe a correlation between homing success and androgen levels, we found three associations between androgen levels and spatial behavior.

Higher baseline androgen levels predicted more exploration after translocation in *D. tinctorius*, and the amount of exploration during the navigation task was associated with an increase in androgen levels in *A. femoralis*. We also found that males, on average, had higher androgen levels and higher exploration rates despite the variation in the parental sex roles and high interindividual variation. Exploration underlies the development of most spatiocognitive abilities (*McNaughton et al., 2006*; *O'keefe and Nadel, 1978*), including spatial memory, presumably used by poison frogs for navigation (*Beck et al., 2017*; *Liu et al., 2016*; *Liu et al., 2019*; *Pašukonis et al., 2014*; *Pašukonis et al., 2016*; *Pašukonis et al., 2019*). Therefore, the association of explorative behavior with androgen levels, especially during the development of spatial memory, might have cascading effects on sex differences in spatial behavior and abilities. Quantifying or manipulating androgen levels during ontogeny and learning, rather than during the spatial task performance, might provide a better understanding of the link between individual differences in the navigational performance observed in our study and androgen levels. We also found that baseline androgens correlated with homing accuracy in *O. sylvatica* in both sexes, further supporting a potential link between androgens and navigational performance. While experimental androgen manipulations are needed to understand the interplay between hormone levels and spatial behavior, some of our findings are in line with the androgen spill-over hypothesis of sex differences in spatial cognition.

## Conclusions

We found that parental behavior drives space use patterns but not navigational performance in poison frogs. Most observed sex differences indicated more developed navigational ability and increased exploratory tendency in males, even in species where females show wider-ranging movement for parental care and mate-seeking. Indeed, most previous literature on sex differences in vertebrate spatial abilities shows either no sex differences or better performance in males. We also found higher average androgen levels in males of all three species despite the marked species differences in parental sex roles and aggressive female behavior. However, there was a high interindividual variation in androgens and a large overlap between sexes. Some of this interindividual variation in androgen levels was related to individual differences in exploration and homing accuracy, suggesting an interplay between androgens and spatial behavior. Therefore, our findings are more consistent with the androgen spillover hypothesis than the widely accepted adaptive specialization hypothesis based on sex difference in home range size or the complexity of spatial behavior. We speculate that sex differences in spatial abilities could be a byproduct of selective pressures on sexual traits such as aggressiveness and the associated increase in androgen levels. However, the indirect effects of androgens,

such as increased male exploration tendency, are likely to be adaptive and maintained in the context of male territoriality and widespread male parental care in poison frogs.

## Materials and methods
### Study species
We studied three poison frog species with different life histories and parental sex roles: the Brilliant-Thighed Poison Frog (*A. femoralis* [Boulenger 1884], Aromobatidae), the Dyeing Poison Frog (*D. tinctorius* [Cuvier 1797], Dendrobatidae), and the Diablito Poison Frog (*O. sylvatica* [Funkhouser 1956], Dendrobatidae). *A. femoralis* and *D. tinctorius* occur in syntopy in the Guiana Shield, but *A. femoralis* has a wider range across Amazonia (*Grant et al., 2006*). *O. sylvatica* is endemic to the Chocoan Rainforest of the Pacific Coast of Ecuador and Colombia (*Grant et al., 2006*). All three species are diurnal, breed throughout the local rainy season, and shuttle tadpoles from terrestrial clutches in the leaf litter to aquatic tadpole nurseries (*Grant et al., 2006*; *Silverstone, 1975*; *Silverstone, 1976*). In *A. femoralis*, groups of up to ~20 tadpoles are predominantly transported by territorial males and deposited in terrestrial pools (*Ringler et al., 2013*; *Roithmair, 1992*). Females take over tadpole transport when males disappear (*Ringler et al., 2015b*). In *D. tinctorius*, males transport one or two tadpoles at a time (but see *Fischer and O'Connell, 2020*; *Rojas and Pašukonis, 2019* for reports of female transport) to terrestrial and arboreal pools (*Fouilloux et al., 2021*; *Rojas, 2014*; *Rojas, 2015*; *Rojas and Pašukonis, 2019*). After tadpole deposition into pools, no further parental care is provided in *A. femoralis* and *D. tinctorius*. In contrast, in *O. sylvatica*, females transport one or two tadpoles at a time and deposit them into water-filled plants (*Silverstone, 1973*; *Summers, 1992*; *Zimmermann and Zimmermann, 1981*). Tadpoles feed on unfertilized eggs, which the mother returns to provide every ~3–7 days (this study and personal observation by E. Tapia communicated to LAC). In all three species, males and females show site fidelity, but the levels of aggressiveness and territoriality vary. In *A. femoralis* and *O. sylvatica*, males vocally advertise and aggressively defend small territories, while females visit males for mating and show no aggressive behavior (*Fischer et al., 2020*; *Ringler et al., 2009*; *Roithmair, 1992*; *Silverstone, 1973*; *Summers, 1992*; this study). In *D. tinctorius*, both males and females show intrasexual aggression as part of territoriality and/or mate guarding, but males do not vocally advertise (*Born et al., 2010*; *Rojas and Pašukonis, 2019*; this study).

### Study sites
Data for *A. femoralis* and *D. tinctorius* were collected over five field seasons between 2016 and 2020 in two different plots at the Nouragues Ecological Research Station (4°02′ N, 52°41′ W) in the Nature Reserve Les Nouragues, French Guiana. One plot consists of ~25 ha of lowland rainforest bordering the Arataï river where both *A. femoralis* and *D. tinctorius* naturally occur. The other plots consist of a 5-ha island in the Arataï river where *A. femoralis* and *D. tinctorius* were absent, but an experimental population of *A. femoralis* was successfully introduced in 2012 (*Ringler et al., 2015a*). The island population relies primarily on an array of artificial pools for breeding but otherwise lives under natural conditions (*Ringler et al., 2018*). Data for *O. sylvatica* were collected at two sites in Ecuador: Sapoparque La Florida (0°15′ S, 79°02′ W) in 2017 and Reserva Canandé (0°32′ N, 79°13′ W) in 2019. The La Florida study area (enclosure site hereafter) consisted of a free-ranging population of *O. sylvatica* introduced and kept in two forest enclosures of ~0.25 ha each inside their natural habitat. The natural breeding pools in the enclosures were supplemented by a high density of small artificial plastic pools and suitable plants. The Reserva Canandé study site (natural site hereafter) consisted of a natural *O. sylvatica* population, relying only on natural pools (water-filled plants). For summarized study site and different dataset information see *Supplementary file 1* (*Table 1i and j*).

### Frog tracking
We used two previously described tracking methods: harmonic direction-finding (HDF) with passive transponders for *A. femoralis* and *O. sylvatica* (for more details see *Beck et al., 2017*; *Fischer et al., 2020*; *Pašukonis et al., 2014*) and radio-tracking with Very High Frequency (VHF) transmitters for the larger *D. tinctorius* (for more details see *Pašukonis et al., 2018*; *Pašukonis et al., 2019*). For HDF, we used a handheld transceiver (R8 and R9, Recco AB, Lindigö, Sweden) to detect miniature transponders (custom-made and commercial transponders R-30CL, Recco AB) attached to frogs with a silicone

waistband. The tags with waistbands weighed ~0.1 g and constituted ~5% of adult *A. femoralis* and *O. sylvatica* body weight. *D. tinctorius* were equipped with miniature VHF transmitters (BD2X, Holohil Systems Ltd, Carp, ON, Canada; NTQ2, Lotek Wireless, Newmarket, ON, Canada; PicoPip, Biotrack, Wareham, UK; V5, Telemetrie-Service Dessau, Dessau-Rosslau, Germany). Frogs were located using a portable radio-tracking receiver (Sika, Biotrack Ltd) and a flexible Yagi-antenna (Biotrack Ltd). The tags weighed 0.35–0.4 g and constituted 6–12% of adult *D. tinctorius* body weight. Frogs were shortly handled every few days to check the tag fit, and the tag was adjusted or removed as necessary. Antennas occasionally tangled and snagged on the vegetation, in which case we intervened to release the frog. Five out of 311 frogs died during the study: one due to experimenter error, one due to an antenna snagged on the vegetation, two due to predation, and one for unknown reasons. Some frogs (16%) experienced skin damage ranging from superficial abrasions to deep open wounds from attachment. The prevalence of wounds varied between species (*A. femoralis* 6%, *O. sylvatica* 18%, *D. tinctorius* 25%) and field seasons. Fast recovery of skin injuries was observed in the frogs that were recaptured after removing the tag.

We tagged and tracked 311 frogs, located each frog multiple times a day (see further below), and recorded its position and any observed behavior. To record the behavior, we tried to visually spot the frog with every recorded position, although this was not always possible due to dense habitat. In these cases, we narrowed the location to approximately 1 m. All study plots were mapped using precision compasses and laser distance meters, establishing a network of labeled reference points (for method details, see *Ringler et al., 2016a*). To map frog movements, we measured distance and direction from these reference points and recorded data on digital maps with the GIS software ArcPad 10 (ESRI, Redlands, USA) on handheld devices (Vanquisher SV-86 rugged Windows tablet, Sinicvision Technology Co, Shenzhen, China; WinTab 9, Odys, Willich, Germany; MobileMapper10; SpectraPrecision, Westminster, CO, USA). Occasionally, when frogs moved out of the mapped area, we recorded their location by averaging at least 30 GPS points with a GPS device MobileMapper 10. All data were collected as GIS spatial points with associated behavioral information, checked point-by-point at least twice, and corrected for errors such as wrong frog identities, duplicates, and impossible locations by one of the experimenters in GIS software QGIS (versions 2.14 and 3.18, *QGIS.org, 2022*) and ArcMap (various versions, ESRI, Redlands, USA). All suspect points where frog identity or location could not be unambiguously confirmed and corrected were removed.

## Quantification of space use

To quantify space use, we tagged 36 *A. femoralis*, 31 *D. tinctorius*, and 83 *O. sylvatica*, which we localized 6682 times and tracked for periods ranging from <1 to 45 days per individual. A subset of these data was used in previous publications on female space use in *A. femoralis* (*Fischer et al., 2020*) and tadpole transport in *D. tinctorius* (*Pašukonis et al., 2019*). When a frog lost its tag, we attempted to recapture and retag the individual to continue tracking. Identity was confirmed based on unique dorsal or ventral coloration patterns. We excluded all frogs that were tracked for less than two full days (not including the tagging and tag removal/loss day) without any manipulation and handling (tag checks or retagging). We also removed short and temporally disconnected tracking periods when the frog was retagged. In the end, we had data to quantify the space use of 29 *A. femoralis* (17 females and 12 males) tracked from 5 to 16 days (median = 14 days) and 26 *D. tinctorius* (11 females and 15 males) tracked for 3–45 days (median = 14 days). We tracked 29 *O. sylvatica* (14 females and 15 males) for 7–20 days (median = 10 days) within enclosures and an additional 37 *O. sylvatica* (20 females and 17 males) for 2–8 days (median = 3 days) at the natural site.

The sampling rate (number of locations per day) varied for different datasets and tracking periods. Movements associated with specific behaviors of interest, such as the tadpole transport, were often sampled at a higher frequency. As the sampling rate influences the spatial parameters, we standardized the data by downsampling all datasets to match the datasets with the lowest sampling rate (~4 points per day). An experienced observer (AP) down sampled the data in a two-step procedure that allowed us to maintain the maximum spatial and behavioral information. We first counted the number of points per day and selected the days with more than 2 points above the daily average for the respective dataset. We then removed redundant points while trying to keep the most spatially and temporally distributed points, and points with rare behaviors, such as parental care and mating. In a second step, we automatically downsampled the remaining dataset to a minimum sampling interval of

60 min while retaining intermediate points for long (>20 m), fast movements, occurring within shorter than 60 min. The resulting dataset had 3–7 points per day per frog (median = 4).

To quantify the space use, we calculated the daily cumulative distance traveled (daily travel hereafter) for 84 frogs tracked for at least two full days. For 76 frogs tracked for a minimum of 7 days, we also quantified the maximum movement extent area (movement extent hereafter) as a minimum convex polygon (MCP) and the home range as 95% utilization density (UD) contour derived from kernel density estimation (KDE) using the 'mcp' and 'kernelUD' functions of the 'adehabitatHR' R package (*Calenge, 2006*). The smoothing parameter for the KDE was calculated with a conservative plug-in bandwidth selection method with the 'hpi' function of the 'ks' R package (*Chacón and Duong, 2018*). Daily travel is a good proxy for the mobility of the animal, but it does not capture the range of the animal's movement. Movement extent calculated as the MCP encompassing all tracked positions is a proxy for the size of the area explored by an animal including the rare excursions, but it tends to overestimate the size of the area familiar to the animal. The home range calculated as an area under 95% UD is a proxy for the area fully familiar to an animal, but it excludes the areas where the animal spends little time, thus underestimating the full extent of a familiar area. For summary of space use variables see *Supplementary file 1* (*Table 1k, l*).

To quantify the influence of parental and reproductive behaviors on movement, we grouped behaviors into three categories: parental and pool associated behavior (parental behavior hereafter), mating associated behavior (mating behavior hereafter), and 'other' when neither parental nor mating behavior was observed. Under parental behavior, we included direct observations of tadpole transport and egg-feeding, as well as all points where the frogs were located within 1 m of known breeding pools. Males of *O. sylvatica* were sometimes observed next to breeding pools, but we did not consider that as parental behavior because parental care has not been reported in male *O. sylvatica* nor observed in this study. Under mating behavior, we included direct observations of courting and mating behaviors, as well as all points where the frogs were located within 1 m from an opposite-sex individual. We categorized each tracking day as 'parental', 'mating', or 'other' whenever the respective behavior was observed at least once on that day. On the days when both parental and mating behavior occurred (12 out of 84 parental behavior days), we categorized the day as 'parental' because parental movements were larger in scale. To evaluate the influence of behavior on movement, we then compared the distance traveled on 'parental', 'mating', and 'other' days.

Our tracking-based space use measures represent a snapshot of an animal's long-term movement, and some species were tracked in experimental study plots confined by enclosure or water. Therefore, we further validated our tracking data described above with three supplementary datasets, including multiyear capture–recapture data in natural populations of *A. femoralis* and *D. tinctorius* (for more details on the capture–recapture method see *Ringler et al., 2009*) and a short-term tracking of *O. sylvatica* in a natural population. For *A. femoralis* and *D. tinctorius*, we compared the sex differences observed in the tracking data to a multiyear capture–recapture data collected at the same study site over 3 (2009–2011 for *D. tinctorius*, B. Rojas, M. Ringler, unpublished data) or 6 years (2014–2019 for *A. femoralis*, M. Ringler, E. Ringler, A. Pašukonis, unpublished data). For this analysis, we only included the individuals that were recaptured at least 30 days apart. For each of these recaptured individuals, we calculated the maximum linear distance between all recapture points. Maximum linear distance is correlated to the movement extent area calculated for tracked individuals, but it can be calculated from as few as two recaptures of an individual. We also calculated the maximum linear distance for each individual based on the tracking data. Contrary to the tracking data for *A. femoralis*, which was collected on an island with artificial pools, *A. femoralis* capture–recapture data were collected from a nearby natural population and thus served as additional validation for the effects of the island and artificial pools on the patterns of space use. For *O. sylvatica*, we checked if the sex differences seen in enclosures were consistent with short-term tracking (1–7 days, median = 3 days) of 37 *O. sylvatica* (17 females and 20 males) data collected at a natural population. For both datasets (enclosures and natural site), we measured the movement extent distance, as the maximum linear distance in meters between all locations of the same individual.

## Quantification of navigational performance

To quantify the navigational performance, we initially tagged 78 *A. femoralis*, 83 *D. tinctorius*, and 52 *O. sylvatica* (same *O. sylvatica* individuals as for space use quantification at a natural population). Out

of these frogs, we carried out translocations and measured homing for 64 *A. femoralis* (32 females and 32 males), 67 *D. tinctorius* (35 females and 32 males), and 39 *O. sylvatica* (19 females and 20 males). Other frogs were not translocated because of dry weather conditions, lost tags, technical difficulties, and various other field constraints. Most frogs were tagged and tracked for at least 24 hr before translocation to establish site fidelity to the tagging areas. We presumed the tagging area to correspond to defended territories or core areas within the home range (collectively termed home areas hereafter). We further confirmed site fidelity by behavioral observations of calling, courtship, and repeated use of shelters. In a few instances, *A. femoralis* were translocated immediately after tagging because the territories were already known from a concurrent study (*Rodríguez et al., 2020*). We did not translocate frogs transporting tadpoles or continuously moving away from the tagging site. The locations of each frog recorded in the home area before the translocation were used as reference points to establish the correct homeward direction and homing success. We translocated one male and one female from nearby home areas simultaneously and released them 50 or 200 m away from their respective home areas. Frog movements were then tracked for 4 or 6 days for 50 and 200 m translocation, respectively, or until the frogs returned within ~10 m from their home areas. Frogs that did not return home within the given time were captured and released back at their home areas. In a few instances, the frogs were tracked for longer than 4 or 6 days, but the trajectories were truncated for the analysis.

For translocation, each frog was captured and placed in an airtight and opaque container. We chose, measured, and located a release site on the digital map of the study area and carried the frogs to the release site. To disorient the frogs, the container was rotated multiple times and never carried in a straight line from the capture to the release site. All frogs were captured and released in the afternoon, and the translocation usually took between 30 and 90 min. We translocated one male and one female simultaneously from the same area toward the same direction and attempted to vary the translocation direction between pairs within the landscape constraints of the field site. Most frogs were released by placing the container on the ground for at least 5 min and then gently opening the lid allowing the frog to leave. In the case of *A. femoralis* tracked in 2017, frogs were removed from a bag by hand and placed under a flowerpot with an opening for an exit. If the frogs did not leave the pot within 30–60 min, we lifted the pot by hand. These periods of 30–60 min were not included in the analysis.

We directly observed and mapped the initial movements of each frog for at least 30 min after release. For this, we set up a radial grid with a 3-m radius at the release site, made of colored strings for visual reference, and mapped the frog movements at approximately 0.3-m precision on a tablet PC running a custom Python (version 2.7.3) script ('Frogger Logger' available on https://osf.io/3bpn6/) allowing us to record the frog position in relation to the visual reference grid. Following the direct observation, we attempted to locate the frogs approximately every 20–60 min, although longer sampling gaps occurred due to local terrain and weather constraints.

For summary of navigation variables see *Supplementary file 1* (*Table 1k, l*). To measure the movement strategy and the navigational performance, we quantified (1) homing success, (2) explored area, (3) homing trajectory straightness, (4) angular deviation from the home direction, and (5) homing duration. (1) We assumed that the frog is showing homing behavior (yes/no) if the frog approached at least 70% of the distance from the release site to the home area center, defined as the geometric average of frog positions prior to translocation. (2) To estimate the explored area, we calculated the total area within 5 m around the movement trajectory. The value is based on a putative perceptual range of a frog being at least 5 m. (3) Trajectory straightness was calculated as the ratio between the straight-line distance from the release site to the end of the homing trajectory and the cumulative distance of the actual homing trajectory. To calculate explored area and trajectory straightness, trajectories were downsampled to the minimum sampling interval of 15 min. Trajectory straightness measures are influenced by the sampling rate, which was variable in our dataset. Therefore, we also quantified homing accuracy as an angular deviation from the home direction at fixed distance thresholds from the release site. The goal was to quantify the accuracy during the initial orientation near the release site, adjusted for translocation distance. (4) We measured the angular deviation from the home center direction at the first point where the frog first crossed a 10 ∓ 5 or 40 ∓ 20 m radius circle drawn around the release site for 50 and 200 m translocation, respectively. This corresponded to ~20% of the total translocation distance for each translocation distance. (5) The homing duration was calculated as the time from the release of the frog to the moment when the frog crossed a 10-m buffer drawn around the home

area polygon. Night-time (12 hr per night) was excluded from homing duration because none of the study species moves at night. Homing success, explored area, and angular deviation were calculated for all translocated frogs. Trajectory straightness and duration were only estimated for the frogs that successfully returned home. Trajectory straightness and angular deviation could not be estimated for some frogs due to missing data in the trajectory.

## Quantification of androgen levels

For a subset of the frogs used in navigation experiments, we quantified androgen levels using non-invasive water sampling, following the methodology described elsewhere (*Baugh and Gray-Gaillard, 2021*; *Rodríguez et al., 2022*). Androgen levels were quantified once in the morning within 2 days before translocation (baseline hereafter) and again in the morning after the frogs returned home by themselves or were returned by the experimenter. We did not quantify androgen levels for the *A. femoralis* and *D. tinctorius* translocated in 2017. Each frog was placed in 40 ml of distilled water inside a small glass container with a dark cover for 60 min at the frog capture location and released immediately after. The water sample was pushed through a C18 cartridge (SPE, Sep-Pak C18 Plus, 360 mg Sorbent, 55–105 µm particle size, #WAT020515, Waters Corp, Milford, MA, USA) with a 20-ml sterile syringe. Cartridges were immediately eluted with 4 ml of 98% EtOH into 5 ml glass vials and were stored at first at 4°C when in the field and then transferred to −20°C until analysis.

To estimate androgen concentration, we used a commercial enzymatic immunoassay for testosterone (ADI-900-065, RRID: AB_2848196, Enzo Life Sciences, Farmingdale, NY, USA). Before the ELISA, 1 or 2 ml of the original 4 ml sample was dried down with N2 at 37°C and resuspended with 250 µl of the assay buffer (provided in the kit), and incubated overnight at 4°C. Samples were brought to room temperature and shaken at 500 rpm for 1 hr prior the assay. Samples were plated in duplicate and assays were performed following the manufacturer's protocol. Plates were read at 405 nm, with correction at 570 nm, using a microplate reader (Synergy H1, BioTek Instruments, Winooski, VT, USA) and the concentration of androgens was calculated using the software Gen5 (version 3.05, BioTek Instruments, Winooski, VT, USA). The lower and upper detection limit for the assay were 7.8 and 2000 pg/ml and samples that fell out of this range were removed from the analysis. The cross-reactivity of the testosterone antibody with other androgens is below 15% according to the manufacturer's manual. Samples with the average intra-assay coefficient of variation (CV) above 15% were excluded from the analysis and the resulting average CV was 5.7%. The average inter-assay CV was 7% for five out of nine assays and not available for the other four due to experimenter error. The final reported concentrations were adjusted for the sample volume taken (1 or 2 ml) and sample concentration during drying-resuspending.

## Weather variables

We measured the understory ambient temperature with temperature-data loggers (HOBO U23 Pro v2, Onset Computer Corp, Bourne, USA) placed ~30 cm above ground and recording temperature at 15 or 30 min intervals. Because our study species are exclusively diurnal, we calculated daytime temperature by averaging all measures from sunrise to sunset. At the *O. sylvatica* enclosure site, we manually measured temperature three times per day at the start, middle, and end of each tracking session with a handheld electronic thermometer (GFTH 95, GHM Messtechnik, Regenstauf, Germany) held slightly above the ground. We averaged the three measurements to obtain daytime temperature. We also calculated mean daytime temperatures for each frog during the navigation experiments by averaging daytime temperatures over the tracking period of each frog (also see *Supplementary file 1*, *Table 1I*). We did not use the rainfall data because they were strongly correlated with daytime temperature and missing for some tracking periods.

## Data analyses

All statistics were generated in R Studio (version 1.0.153, *RStudio Team, 2020*) running R (version 3.6.3, *R Development Core Team, 2020*). Space-use plots were generated in QGIS (version 2.14, *QGIS.org, 2022*), box plots, trajectory plots, and bar plots were generated in R with the 'ggplot2' package (*Wickham, 2016*), circular plots with the 'circular' package (*Agostinelli and Lund, 2022*), and model plots with the 'sjPlot' package (*Lüdecke, 2021*). Schematic representations and further editing were done with Adobe Illustrator (version 25.2.3, Adobe Inc, Mountain View, CA, USA), Inkscape

(version 1.0.2, Inkscape Project 2020), and Microsoft PowerPoint (version 16.54, Microsoft Corp, Redmont, WA, USA). All variables and statistical models are summarized in *Supplementary file 1* (*Table 1k–m*).

## Space use

Spatial variables (daily travel, movement extent area, movement extent distance, home range, and explored area) approximately followed a log-normal distribution and thus we transformed the raw data using the natural logarithm to fit the model assumptions. Before all other analyses, we evaluated if tagging-related skin injuries affected frog mobility. We categorized the injuries into three categories of severity: no injury, superficial abrasion, and skin lesion. We then included injury as a categorical predictor in models of home range size, movement extent, and daily travel. Injuries had a significant effect on home range size and daily travel in *A. femoralis*, but not in *D. tinctorius* and *O. sylvatica*. Based on these results, we excluded *A. femoralis* (one male and one female) with skin lesions from space use analyses. The overall results of all models remained the same with and without injured frogs, but we report the conservative values after excluding the two injured frogs.

To investigate sex differences in movement extent and home range, we fitted two linear models (LMs) with movement extent and home range as responses in each model, respectively, and sex, species, and their interaction as predictors. As there was a strong interaction between species and sex for both models, we fitted a separate LM for each species with sex as the predictor. As movement extent and home range often correlate with tracking duration, we included the number of days tracked as a covariate. To investigate the influence of behavior and sex on the daily movement, we fitted linear mixed-effects models (LMMs) with daily travel as the response variable using the 'lmer' function within the 'lme4' R package (*Bates et al., 2015*). We first fitted an LMM for each species with sex, behavior, and daytime temperature as fixed factors and frog identity and tracking date as random factors. We checked for a correlation between daily travel and number of points per day for each species (significant for *D. tinctorius* and *O. sylvatica*, but not for *A. femoralis*) and used number of points per day as a covariate in LMMs when the correlation was significant. For model selection we followed an information-theoretic approach (*Burnham and Anderson, 2002*) based on the Corrected Akaike's Information Criterion (AICc). We calculated models with all combinations of fixed factors (behavior, sex, daytime temperature) while keeping random factors (id and date) and the covariate (number of points per day). We selected the best single model using the 'model.sel' function within the 'MuMIn' R package (*Bartón and Barton, 2020*). Because parental behaviors mostly occurred in one sex, we also analyzed the influence of behavior on movement for each sex separately. We fitted separate LMMs for each sex of each species with behavior as a fixed factor (two or three levels: parental, mating, and other as appropriate for each species and sex), and frog identity and tracking date as random factors. The daytime temperature was only included as a fixed factor if the best model based on AICc included temperature (it did for *A. femoralis*, but not for *D. tinctorius* and *O. sylvatica*). We used least-squares means contrasts to compare daily travel between behavioral categories. Post hoc comparisons were done with the 'emmeans' function within the emmeans R package (*Lenth et al., 2022*) with p values adjusted by Tukey's method for multiple comparisons (*Tukey, 1977*). To compare the tracking-based space-use data for *A. femoralis* and *D. tinctorius* with long-term data, we fitted a separate LM for each species with capture–recapture-based movement extent distance as the response variable and sex and time period between recapture points as predictors. We also fitted a separate LM for each species with tracking-based movement extent distance as the response variable and sex as predictor. To validate the space-use data for *O. sylvatica*, we fitted a separate LM for natural site and enclosure data with movement extent distance as the response variable and sex and tracking duration as predictors. We also fitted an LMM for the natural site and enclosure data with daily travel as the response variable, sex as a fixed factor, and frog identity as a random factor.

## Navigational performance

As species and frogs translocated to different distances showed qualitatively very different movement and homing patterns, we fitted separate models per species and per translocation distance. Before all other analyses, we evaluated if tagging-related skin injuries affected frog mobility following the procedure described above. We included injury as a categorical predictor in models of explored area and homing duration. The injury had a significant effect in the model of explored area size in *O. sylvatica*

translocated 200 m. Based on these results, we excluded *O. sylvatica* (three males and three females) with skin lesions from navigation analyses. The overall results of all models remained the same with and without injured frogs, but we report the conservative values after excluding injured individuals.

To investigate sex differences in the homing success we fitted generalized linear models (GLMs) with binomial error distribution and Logit link function using the 'glm' function within R package stats (*R Development Core Team, 2020*). We used the homing success as a binary response variable with sex, frog weight (except for *O. sylvatica*), and the mean daytime temperature during the entire tracking period as predictors. To investigate sex differences in exploration, we fitted an LM with the natural log-transformed explored area as the response variable and sex, mean daytime temperature, and frog weight as predictors. To investigate sex differences in homing trajectory straightness we fitted a beta regression model for proportions via maximum likelihood using the 'betareg' function within the R package betareg (*Cribari-Neto and Zeileis, 2010*). We used straightness (ratio between 0 and 1) as the response variable with sex and mean daytime temperature as predictors. We also evaluated if each group (males and females after 50 or 200 m translocation) were significantly orientated toward home at the measured thresholds with the Rayleigh test of circular uniformity with a specified mean home direction as an alternative hypothesis using the 'rayleigh.test' function within the 'circular' R package. We then compared the circular distributions of males and females for each species and translocation group with an MANOVA based on trigonometric functions (*Landler et al., 2021*) using the 'manova' functions within the 'stats' R package. To investigate sex difference in homing duration we fitted an LM with natural log-transformed homing duration as the response variable and sex, mean daytime temperature, and frog weight as predictors.

## Androgen levels and navigation

To investigate the relationship between androgens, sex, and movement, we fitted a series of separate models for each species. We first fitted LMMs with log-transformed androgen levels as the response variable, sex, and time point (two levels: baseline or back home) as fixed factors, and frog identity as a random factor. For all three species, there was no interaction between sex and sampling time point, and the interaction factor was excluded. To investigate the relationship between baseline androgens and spatial behavior during navigation, we first fitted an LM for each species with the log-transformed explored area as the response variable and baseline androgen levels, sex, translocation distance, homing success, mean daytime temperature, and frog weight as predictors. For successfully homing frogs, we also fitted two GLMs (Gamma distribution, inverse link function) with trajectory straightness and homing duration as response variables and baseline androgen levels, sex, translocation distance, mean ambient temperature, and frog weight as predictors. To be able to evaluate the relative influence of different fixed factors on the response variables, we standardized all continuous predictors by centering and scaling using the 'stdize' function from the 'MuMIn' R package. We did not include sex for GLMs of trajectory straightness and homing duration of *A. femoralis* because androgen levels were available for only one homing female. If movement itself had an influence on androgen levels, we expected the amount of exploration to have a significant influence on androgen levels after the navigation task. Therefore, we calculated delta androgen levels by subtracting the baseline from the back home levels and fitted an LM with delta androgen levels as a response variable and the explored area, sex, tracking duration, homing success, mean daytime temperature, and frog weight as fixed factors. To reduce the number of covariates, for each model mentioned above, we compared the full models against a model without the mean temperature or the frog weight using the 'drop1' function of the stats R package. We removed mean temperature and frog weight from the final model if the model excluding these factors (separately) was not significantly different ($p > 0.1$).

## Ethics statement and permits

We strictly adhered to the current law in the USA, France, Ecuador, and the European Union, and followed the 'Guidelines for use of live amphibians and reptiles in the field and laboratory research' by the Herpetological Animal Care and Use Committee (HACC) of the American Society of Ichthyologists and Herpetologists (*Beaupre et al., 2004*) and the Association's for the Study of Animal Behaviour (ASAB) 'Guidelines for the use of live animals in teaching and research' (*ASAB, 2020*). The research was approved by the Institutional Animal Care and Use Committee of Stanford University (protocol ID 33211, issued to LAO) and by the Animal Ethics and Experimentation Board of the Faculty of Life

Sciences, University of Vienna (approvals no. 2016-002, 2016-003, 2018-10, issued to AP). The permits in French Guiana were issued by the local authorities (DIREN permits: R03-2016-10-21-002, no. 2015-289-0021, no. 2011-44/DEAL/SMNBSP', issued to ER). Sample collection in French Guiana was declared following the implementation of the Nagoya Protocol on access to genetic resources and the fair and equitable sharing of benefits arising from their utilization (APA permit TREL2002508S/307). In addition, all protocols for fieldwork were approved by the scientific committee of the Nouragues Ecological Research Station (approval communicated to AP) and the Nouragues Nature Reserve (partnership agreement no. 01-2019 with AP, BR, ER, and MR). The permits in Ecuador were issued by the local authorities (Ministerio de Ambiente, approval document no. 013-18-IC-FAU-DNB/MA, issued to LAC). In addition, the authorization to work in Reserva Canandé was given by reserve authority Fundación Jocotoco, Ecuador (approval communicated to AP).

## Acknowledgements

We are grateful to the staff of the CNRS Nouragues Ecological Research Station, Nouragues Nature Reserve, Fundación de Conservación Jocotoco, Reserva Canandé, Centro Jambatu de Investigación y Conservación de Anfibios, and Wikiri for logistic support and generously providing access to the study sites. We are deeply grateful to these organizations and their dedicated staff for their commitment to preserving our natural world and facilitating research. We thank Kristina Beck, Steffen Weinlein, Susanne Stückler, Eva K Fischer, Italo Tapia, and Vincent Premel for assistance in the field, Jinook Oh for coding the 'Frogger Logger' Python script, George Perry, Edward Clint, and one anonymous reviewer for constructive feedback on the manuscript, and Walter Hödl for continuing inspiration and mentorship that has led to this study. This work is part of a partnership agreement between AP, BR, MR, ER, and the Nouragues Nature Reserve to improve and spread the knowledge about amphibians.

The study was funded by The European Commission's Horizon 2020 research and innovation program under the Marie Sklodowska-Curie Actions grant agreement no. 835530 to AP, LAO, and Simon Benhamou; National Science Foundation CAREER grant (IOS-1845651) to LAO; Association for the Study of Animal Behaviour (ASAB) 2016 Research Grant to AP; Austrian Science Fund (FWF): P24788 and P31518 and FWF Herta-Firnberg Grant T699 to ER; P33728 and FWF Erwin-Schrödinger Fellowship J3868-B29 to MR; FWF Erwin-Schrödinger Fellowship J3827-B29 to AP; CNRS Nouragues Travel Grants (AnaEE France ANR-11-INBS-0001) NTG2009 and NTG2010 to BR; NTG2015 to AP and BR. LAO is a New York Stem Cell Foundation – Robertson Investigator. ML received funding by the European Union's Horizon 2020 research and innovation program under the Marie Skłodowska-Curie grant agreement no. 79809. BR received funding from the Academy of Finland Research Fellowship (no. 319949). CR was funded by FWF-DK project W-1262 (Speaker: Tecumseh Fitch). LAC acknowledges the support of Wikiri and the Saint Luis Zoo. The Nouragues Ecological Research Station, managed by CNRS, benefits from 'Investissement d'Avenir' grants managed by the Agence Nationale de la Recherche (AnaEE France ANR-11-INBS-0001; Labex CEBA ANR-10-LABX-25-01).

## Additional information

### Competing interests

Lauren A O'Connell: Reviewing editor, *eLife*. The other authors declare that no competing interests exist.

### Funding

| Funder | Grant reference number | Author |
|---|---|---|
| Horizon 2020 Framework Programme | 835530 | Andrius Pašukonis |
| National Science Foundation | IOS-1845651 | Lauren A O'Connell |
| Association for the Study of Animal Behaviour | 2016 Research Grant | Andrius Pašukonis |

| Funder | Grant reference number | Author |
| --- | --- | --- |
| Austrian Science Fund | P24788 | Eva Ringler |
| Austrian Science Fund | P31518 | Eva Ringler |
| Austrian Science Fund | T699 | Eva Ringler |
| Austrian Science Fund | J3868-B29 | Max Ringler |
| Austrian Science Fund | J3827-B29 | Andrius Pašukonis |
| Analyses et Expérimentations pour les Ecosystèm | ANR-11-INBS-0001 | Andrius Pašukonis<br>Shirley Jennifer Serrano-Rojas<br>Marie-Therese Fischer<br>Daniel A Shaykevich<br>Matthias-Claudio Loretto<br>Bibiana Rojas<br>Max Ringler<br>Eva Ringler<br>Camilo Rodríguez |
| New York Stem Cell Foundation | Robertson Investigator | Lauren A O'Connell |
| Horizon 2020 Framework Programme | 79809 | Matthias-Claudio Loretto |
| Academy of Finland | 319949 | Bibiana Rojas |
| Austrian Science Fund | W-1262 | Camilo Rodríguez |
| Saint Luis Zoo | | Luis A Coloma |
| Wikiri | | Luis A Coloma |
| Agence Nationale de la Recherche | ANR-10-LABX-25-01 | Andrius Pašukonis<br>Shirley Jennifer Serrano-Rojas<br>Marie-Therese Fischer<br>Daniel A Shaykevich<br>Matthias-Claudio Loretto<br>Bibiana Rojas<br>Max Ringler<br>Eva Ringler<br>Camilo Rodríguez |
| Austrian Science Fund | P33728 | Max Ringler |

The funders had no role in study design, data collection, and interpretation, or the decision to submit the work for publication.

## Author contributions

Andrius Pašukonis, Conceptualization, Resources, Data curation, Formal analysis, Supervision, Funding acquisition, Validation, Investigation, Visualization, Methodology, Writing – original draft, Project administration, Writing – review and editing; Shirley Jennifer Serrano-Rojas, Data curation, Formal analysis, Investigation, Writing – review and editing; Marie-Therese Fischer, Daniel A Shaykevich, Data curation, Investigation, Writing – review and editing; Matthias-Claudio Loretto, Formal analysis, Funding acquisition, Investigation, Methodology, Writing – review and editing; Bibiana Rojas, Data curation, Funding acquisition, Investigation, Methodology, Writing – review and editing; Max Ringler, Resources, Data curation, Funding acquisition, Investigation, Methodology, Writing – review and editing; Alexandre B Roland, Investigation, Writing – review and editing; Alejandro Marcillo-Lara, Investigation; Eva Ringler, Resources, Supervision, Funding acquisition, Writing – review and editing; Camilo Rodríguez, Formal analysis, Investigation, Methodology, Writing – review and editing; Luis A Coloma, Resources, Writing – review and editing; Lauren A O'Connell, Conceptualization, Resources, Supervision, Funding acquisition, Visualization, Methodology, Writing – original draft, Project administration, Writing – review and editing

## Author ORCIDs

Andrius Pašukonis http://orcid.org/0000-0002-5742-8222

Shirley Jennifer Serrano-Rojas  http://orcid.org/0000-0001-6811-8265
Marie-Therese Fischer  http://orcid.org/0000-0002-6693-8662
Matthias-Claudio Loretto  http://orcid.org/0000-0002-1940-3470
Daniel A Shaykevich  http://orcid.org/0000-0003-3850-0986
Bibiana Rojas  http://orcid.org/0000-0002-6715-7294
Max Ringler  http://orcid.org/0000-0002-4530-4919
Alexandre B Roland  http://orcid.org/0000-0002-9463-9838
Eva Ringler  http://orcid.org/0000-0003-3273-6568
Camilo Rodríguez  http://orcid.org/0000-0002-9748-1773
Lauren A O'Connell  http://orcid.org/0000-0002-2706-4077

### Ethics

We strictly adhered to the current law in the US, France, Ecuador, and the European Union, and followed the 'Guidelines for use of live amphibians and reptiles in the field and laboratory research' by the Herpetological Animal Care and Use Committee (HACC) of the American Society of Ichthyologists and Herpetologists (Beaupre et al., 2004); and the Association's for the Study of Animal Behaviour (ASAB) 'Guidelines for the use of live animals in teaching and research' (ASAB, 2020). The research was approved by the Institutional Animal Care and Use Committee of Stanford University (protocol ID 33211, issued to LAO) and by the Animal Ethics and Experimentation Board of the Faculty of Life Sciences, University of Vienna (approvals no. 2016-002, 2016-003, 2018-10, issued to AP). The permits in French Guiana were issued by the local authorities (DIREN permits: R03-2016-10-21-002, no. 2015-289-0021, no. 2011-44/DEAL/SMNBSP, issued to ER). Sample collection in French Guiana was declared following the implementation of the Nagoya Protocol on access to genetic resources and the fair and equitable sharing of benefits arising from their utilization (APA permit TREL2002508S/307). In addition, all protocols for fieldwork were approved by the scientific committee of the Nouragues Ecological Research Station (approval communicated to AP) and the Nouragues Nature Reserve (partnership agreement no. 01-2019 with AP, BR, ER, and MR). The permits in Ecuador were issued by the local authorities (Ministerio de Ambiente, approval document no. 013-18-IC-FAU-DNB/MA, issued to LAC). In addition, the authorization to work in Reserva Canandé was given by reserve authority Fundación Jocotoco, Ecuador (approval communicated to AP).

### Decision letter and Author response

Decision letter https://doi.org/10.7554/eLife.80483.sa1
Author response https://doi.org/10.7554/eLife.80483.sa2

## Additional files

### Supplementary files
- MDAR checklist
- Supplementary file 1. Supplementary statistics and methods tables.

### Data availability

All data and associated scripts are available as a project on the Open Science Framework platform (https://osf.io/3bpn6/, https://doi.org/10.17605/OSF.IO/3BPN6).

The following dataset was generated:

| Author(s) | Year | Dataset title | Dataset URL | Database and Identifier |
|---|---|---|---|---|
| Pašukonis A, Serrano-Rojas SJ, Fischer MT, Loretto MC, Shaykevich DA, Rojas B, Ringler M, Roland AB, Marcillo-Lara A, Ringler E, Rodríguez C, Coloma LA, O'Connell LA | 2022 | Data and code for "Contrasting parental roles shape sex differences in poison frog space use but not navigational performance." | https://doi.org/10.17605/OSF.IO/3BPN6 | Open Science Framework, 10.17605/OSF.IO/3BPN6 |

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
