## [Editor Report]

In this important paper, the authors use intensive field monitoring and experimentally induced navigational challenges in three species of poison frog to examine two different hypotheses for sex differences in spatial ability. The study is simultaneously rich and complex; the results are solidly consistent with (but not necessarily definitively or exclusively in support of) the hypothesis that androgens may inadvertently affect spatial ability. This paper is of interest to organismal biologists and evolutionary scientists who study cognitive and behavioral sex differences including those with interests in the evolution of complex spatial behaviors.

---

## [Decision Letter]

**Decision letter after peer review:**

Thank you for submitting your article "Contrasting parental roles shape sex differences in poison frog space use but not navigational performance" for consideration by *eLife*. Your article has been reviewed by 2 peer reviewers, and the evaluation has been overseen by George Perry as the Reviewing and Senior Editor. The following individual involved in the review of your submission has agreed to reveal their identity: Edward Clint (Reviewer #2).

Essential revisions:

The reviewers and I are unanimously enthusiastic about the multi-species natural-experimental setup of your study and the data arising from it, which we find makes a substantial contribution to understanding the potential role of sex differences in complex spatial behavior. Yet there is no uniformity in the extent to which we find that the conclusions made in the current version of the paper and particularly in the strength/confidence with which they are made are sufficiently supported (we all agree that some adjustment is necessary but with different views re: the extent). Ultimately, our consensus and my expectation are that – in addition to the other essential revision points and also reviewing and considering whether and how best to address each of the specific points made in the individual reviews, below – in your revision you should relatively aggressively temper the conclusions to recognize limitations of the study with respect to the hypothesis testing framework, while still celebrating the experiment itself and the contribution of these results to advancing our understanding of these complex issues.

1. Consider and explicitly address (and temper conclusions) accordingly how some of the variables of interest in your comparisons may be conflated with other factors that could also impact these behaviors.

2. Develop a robust discussion on the prior work in cowbirds (Guigueno et al., 2014) in the context of the present study, in terms of what the results from both the prior and present studies do and do not reveal with respect to the central hypotheses of interest.

3. On the androgen hypothesis, we are in agreement that your results, while both intriguing and important, are more equivocal than currently presented in your paper. A shift to focus on how the results contribute to our understanding of the degree to which explanation may contribute in a given setting, but without necessarily aiming to conclusively distinguish between the two hypotheses (which are not necessarily mutually exclusive) at this time, is more appropriate given the design and results of the study.

4. (this one is the opposite of an essential revision). You do not need to address the first point in the “Recommendation to Authors” from Reviewer #2 about the statistical analyses, at least in terms of the analyses themselves (you may choose to enhance your discussion on this topic, though it is not required). It is definitely complex to consider how to ideally perform formal hypothesis testing in a multi-species setup. On further consideration, reviewer #2 concluded that your present treatment is sufficient, and I concur.

*Reviewer #1 (Recommendations for the authors):*

This is a very well-written manuscript, which is particularly appreciated by this reviewer given the complexity of the data.

I am concerned that the study suffers from what is sometimes called “statistical hocus pocus,” a situation in which p-values are compared across analyses rather than including all factors in one model. For example, the authors test for the effect of sex separately in the three species. They find a low p-value for some species, but not others and they then conclude that the sex difference itself differs across species (e.g., this is what is depicted in summary figure 5). Most statisticians would point out that this conclusion is not supported by the p values for individual sex differences. The authors should seriously consider a different approach and/or acknowledge this shortcoming; I recognize that putting all species in one model may be complex. But, if the pattern of sex difference across species is the focus of the study, the authors need to use appropriate statistics.

Figure 5: I have a couple of questions about this figure. What data are used to summarize navigation in the “observed” panel? Probability of successful homing? Or efficiency of homing? I presume this is coming from data in figure 2. What is the basis for the “prediction”? The adaptive specialization hypothesis or the androgen spillover hypothesis? The two hypotheses have the same prediction for the two species but only differ for O. sylvatica. In any case, I suggest that the authors clarify in the figure legend.

*Reviewer #2 (Recommendations for the authors):*

I commend the authors for their excellent work. The sophisticated grasp of the subject matter, the clarity and legibility of the prose, the elegance of the figures, and the prudent choice of methods. My critical remarks should be construed as relatively minor issues that I hope contribute to the quality of the manuscript.

1. Empirical data that is contrary to the adaptive specialization hypothesis. Until about 2016, nearly all human sex differences data was based on western samples, painting a potentially misleading WEIRD (in the Joseph Henrich sense) picture of humanity. There is now significant published data that seems to confirm that the western sample data is indeed not representative of humanity. In small-scale societies where men & women travel in similar ways and degrees, the sex difference in spatial ability shrinks or vanishes. To quote Trumble et al 2016:

"These data add to a growing body of evidence suggesting that sex differences in navigational ability noted in many industrial populations may not be universal in subsistence populations (Berry 1966; Cashdan et al. 2012). More acculturated groups with less female foraging (e.g., the Temne, or the settled Hadza in Mangola) show male-biased advantages, whereas populations whose females engage in more extensive foraging (e.g., mobile Hadza, Inuit, Tsimane) do not show sex differences in navigational ability."

Due to the proximate effects of administered androgens and the early age onset of sex differences in spatial ability in humans, it remains possible that androgen spillover affects such dimorphism. However, lifestyle and ranging habits appear to loom far larger. Here are some human literature citations that should be part of the discussion of the human literature:

Jang, H., Boesch, C., Mundry, R., Kandza, V., & Janmaat, K. R. L. (2019). Sun, age and test location affect spatial orientation in human foragers in rainforests. Proceedings of the Royal Society B: Biological Sciences, 286(1907). https://doi.org/10.1098/rspb.2019.0934

Trumble, B. C., Gaulin, S. J. C., Dunbar, M. D., Kaplan, H., & Gurven, M. (2016). No Sex or Age Difference in Dead-Reckoning Ability among Tsimane Forager-Horticulturalists. Human Nature, 27(1), 51-67. https://doi.org/10.1007/s12110-015-9246-3

Berry, J. W. (1966). Temne and Eskimo Perceptual Skills. International Journal of Psychology, 1(3), 207-229. https://doi.org/10.1080/00207596608247156

Cashdan, E., Marlowe, F. W., Crittenden, A., Porter, C., & Wood, B. M. (2012). Sex differences in spatial cognition among Hadza foragers. Evolution and Human Behavior, 33(4), 274-284. https://doi.org/10.1016/j.evolhumbehav.2011.10.005

There are also a couple of fish studies wherein the data can't easily be brought to accord with the adaptive specialization hypothesis. In the freshwater blenny where females have well-documented larger ranges, females nonetheless acquired a spatial ability task far less frequently. In another study of cichlids, more females than males acquired the task and corrected errors faster than males, even though males are territorial and thus do more ranging. See:

Fabre, N., García-Galea, E., & Vinyoles, D. (2014). Spatial learning based on visual landmarks in the freshwater blenny Salaria fluviatilis (Asso, 1801). Learning and Motivation, 48, 47-54. https://doi.org/10.1016/j.lmot.2014.10.002

Wallace, K. J., & Hofmann, H. A. (2021). Equal performance but distinct behaviors: sex differences in a novel object recognition task and spatial maze in a highly social cichlid fish. Animal Cognition, 24(5), 1057-1073. https://doi.org/10.1007/s10071-021-01498-0

2. You make use of both the tag tracking home range data and the capture-recapture data (line 194, 198). Some comments here would be helpful in addressing why these measures may diverge while both remaining useful and appropriate to the analysis. This would help explain the measures for your study but also can help inform future study designs.

---

## [Author Response]

hEssential revisions:The reviewers and I are unanimously enthusiastic about the multi-species natural-experimental setup of your study and the data arising from it, which we find makes a substantial contribution to understanding the potential role of sex differences in complex spatial behavior. Yet there is no uniformity in the extent to which we find that the conclusions made in the current version of the paper and particularly in the strength/confidence with which they are made are sufficiently supported (we all agree that some adjustment is necessary but with different views re: the extent). Ultimately, our consensus and my expectation are that – in addition to the other essential revision points and also reviewing and considering whether and how best to address each of the specific points made in the individual reviews, below – in your revision you should relatively aggressively temper the conclusions to recognize limitations of the study with respect to the hypothesis testing framework, while still celebrating the experiment itself and the contribution of these results to advancing our understanding of these complex issues.1. Consider and explicitly address (and temper conclusions) accordingly how some of the variables of interest in your comparisons may be conflated with other factors that could also impact these behaviors.

We rephrased and tempered the language and conclusions to avoid any categorical statements in favor or against the presented hypotheses. We significantly expanded the discussion on the methodological issues and limitations of our study. Two new discussion paragraphs and restructuring of other were included to address the concerns.

2. Develop a robust discussion on the prior work in cowbirds (Guigueno et al., 2014) in the context of the present study, in terms of what the results from both the prior and present studies do and do not reveal with respect to the central hypotheses of interest.

We considerably broadened the literature scope primarily by restructuring and expanding the introduction. We included five additional human, five cowbird, and three fish studies to highlight the broader context of the adaptive specialization hypothesis and the conflicting evidence in support of it.

3. On the androgen hypothesis, we are in agreement that your results, while both intriguing and important, are more equivocal than currently presented in your paper. A shift to focus on how the results contribute to our understanding of the degree to which explanation may contribute in a given setting, but without necessarily aiming to conclusively distinguish between the two hypotheses (which are not necessarily mutually exclusive) at this time, is more appropriate given the design and results of the study.

Wherever possible, we slightly reworded the phrasing to avoid any categorical statements in favor or against one of the hypotheses. We also expanded the discussion on the limitations of our approach, how our results might be brought in better accord with the adaptive explanations, and how the future studied might better test the adaptive specialization hypothesis against the alternative explanations. Regarding the androgen spill-over hypothesis, we made specific effort to highlight the limitations of our descriptive approach in the original version (lines 664 – 671) and avoid any exaggerated claims in favor the androgen spill-over hypothesis. The strongest claim we previously made was: “Therefore, our findings are more consistent with the androgen spillover hypothesis than the widely accepted adaptive specialization hypothesis.” We believe that this conclusion is supported by our findings, and we expanded the statement to make it slightly more specific. The current phrasing reads:

“Therefore, our findings are more consistent with the androgen spillover hypothesis than the widely accepted adaptive specialization hypothesis based on sex difference in home range size or the complexity of spatial behavior.”

We are open to any additional suggestions on how further nuance the discussion and keep it relative concise.

4. (this one is the opposite of an essential revision). You do not need to address the first point in the "Recommendation to Authors" from Reviewer #2 about the statistical analyses, at least in terms of the analyses themselves (you may choose to enhance your discussion on this topic, though it is not required). It is definitely complex to consider how to ideally perform formal hypothesis testing in a multi-species setup. On further consideration, reviewer #2 concluded that your present treatment is sufficient, and I concur.

We agree that there is a multitude of possible statical approaches to analyzing a complex dataset such as ours. We believe that our main findings are both quantitatively and qualitative clear and the interpretation of the results will not be influenced by applying a different approach, therefore we also consider the present treatment sufficient. We do make an effort to clearly separate the qualitative comparison of species differences from statistically supported quantitative comparisons of sex difference within each species. All data and statistical analyses are now publicly available via the Open Science Framework platform (https://osf.io/3bpn6/, DOI: 10.17605/OSF.IO/3BPN6) for further scrutiny and future reevaluation.

Reviewer #1 (Recommendations for the authors):This is a very well-written manuscript, which is particularly appreciated by this reviewer given the complexity of the data.

We really appreciate the positive feedback and taking into account the challenges of collecting comparative data in the field.

I am concerned that the study suffers from what is sometimes called "statistical hocus pocus," a situation in which p-values are compared across analyses rather than including all factors in one model. For example, the authors test for the effect of sex separately in the three species. They find a low p-value for some species, but not others and they then conclude that the sex difference itself differs across species (e.g., this is what is depicted in summary figure 5). Most statisticians would point out that this conclusion is not supported by the p values for individual sex differences. The authors should seriously consider a different approach and/or acknowledge this shortcoming; I recognize that putting all species in one model may be complex. But, if the pattern of sex difference across species is the focus of the study, the authors need to use appropriate statistics.

This point has been addressed above. We would like to further add that for the space use data, which was more homogenous, we did first present a model with all three species included. The strong interaction effect between sex and species clearly indicates that the sex differences vary between species, but it also obscures the specific patterns and factors influencing sex differences in each species, which we then analyze separately. For the navigation results, because of strong qualitative differences in behavior, which you correctly pointed out, it was not informative to combine species in a single model and we only discuss the obvious qualitative differences when comparing the species.

Figure 5: I have a couple of questions about this figure. What data are used to summarize navigation in the "observed" panel? Probability of successful homing? Or efficiency of homing? I presume this is coming from data in figure 2. What is the basis for the "prediction"? The adaptive specialization hypothesis or the androgen spillover hypothesis? The two hypotheses have the same prediction for the two species but only differ for O. sylvatica. In any case, I suggest that the authors clarify in the figure legend.

We uploaded a new version of the Figure 5 and clarified the legend to address your point. The new version clearly states the predictions are based on the life-history traits (the adaptive specialization based on them), which are summarized in the left panel of the figure. We clarified that the representation of the observed differences is based on homing success/ability rather than on the specific measures of accuracy. We also modified the iconography representing the observed sex difference to indicate that in some conditions a smaller proportion of one sex returns home.

Reviewer #2 (Recommendations for the authors):I commend the authors for their excellent work. The sophisticated grasp of the subject matter, the clarity and legibility of the prose, the elegance of the figures, and the prudent choice of methods. My critical remarks should be construed as relatively minor issues that I hope contribute to the quality of the manuscript.1. Empirical data that is contrary to the adaptive specialization hypothesis. Until about 2016, nearly all human sex differences data was based on western samples, painting a potentially misleading WEIRD (in the Joseph Henrich sense) picture of humanity. There is now significant published data that seems to confirm that the western sample data is indeed not representative of humanity. In small-scale societies where men & women travel in similar ways and degrees, the sex difference in spatial ability shrinks or vanishes. […] There are also a couple of fish studies wherein the data can't easily be brought to accord with the adaptive specialization hypothesis. In the freshwater blenny where females have well-documented larger ranges, females nonetheless acquired a spatial ability task far less frequently. In another study of cichlids, more females than males acquired the task and corrected errors faster than males, even though males are territorial and thus do more ranging.

Based on your suggestions, we included five additional human, five cowbird, and three fish studies to present the broader context of the adaptive specialization hypothesis and the conflicting evidence in support of it. While we found that not all suggested references are necessarily in contrary to the adaptive specialization hypotheses, the hope the restructured and expanded introduction provides more nuance and context.

2. You make use of both the tag tracking home range data and the capture-recapture data (line 194, 198). Some comments here would be helpful in addressing why these measures may diverge while both remaining useful and appropriate to the analysis. This would help explain the measures for your study but also can help inform future study designs.

We included additional information about the different space use measures in the methods and a full paragraph in the discussion.